# Functional and epigenetic phenotypes of humans and mice with DNMT3A Overgrowth Syndrome

Amanda M. Smith[1], Taylor A. LaValle[1], Marwan Shinawi [2], Sai M. Ramakrishnan[1], Haley J. Abel[1], Cheryl A. Hill [3], Nicole M. Kirkland[3], Michael P. Rettig [1], Nichole M. Helton[1], Sharon E. Heath[1], Francesca Ferraro[1], David Y. Chen [4], Sangeeta Adak[5], Clay F. Semenkovich [5], Diana L. Christian[6], Jenna R. Martin[6], Harrison W. Gabel[6], Christopher A. Miller[1] & Timothy J. Ley [1✉]

Germline pathogenic variants in *DNMT3A* were recently described in patients with overgrowth, obesity, behavioral, and learning difficulties (*DNMT3A* Overgrowth Syndrome/DOS). Somatic mutations in the *DNMT3A* gene are also the most common cause of clonal hematopoiesis, and can initiate acute myeloid leukemia (AML). Using whole genome bisulfite sequencing, we studied DNA methylation in peripheral blood cells of 11 DOS patients and found a focal, canonical hypomethylation phenotype, which is most severe with the dominant negative *DNMT3A*[R882H] mutation. A germline mouse model expressing the homologous *Dnmt3a*[R878H] mutation phenocopies most aspects of the human DOS syndrome, including the methylation phenotype and an increased incidence of spontaneous hematopoietic malignancies, suggesting that all aspects of this syndrome are caused by this mutation.

[1] Division of Oncology, Section of Stem Cell Biology, Department of Internal Medicine, Washington University School of Medicine, St. Louis, MO, USA. [2] Department of Pediatrics, Division of Genetics and Genomic Medicine, Washington University School of Medicine, St. Louis, MO, USA. [3] Department of Pathology and Anatomical Science, University of Missouri School of Medicine, Columbia, MO, USA. [4] Division of Dermatology, Department of Internal Medicine, Washington University School of Medicine, St. Louis, MO, USA. [5] Division of Endocrinology, Metabolism & Lipid Research, Department of Internal Medicine, Washington University School of Medicine, St. Louis, MO, USA. [6] Department of Neuroscience, Washington University School of Medicine, St. Louis, MO, USA. ✉email: timley@wustl.edu

O vergrowth syndromes are a heterogeneous group of rare disorders that are characterized by a global or localized tissue hypertrophy; the genetic causes of these syndromes are emerging as more patients' genomes are molecularly scrutinized. The first report of *DNMT3A* overgrowth syndrome (DOS, also called Tatton–Brown–Rahman Syndrome; TBRS; MIM 615879) described a syndrome of increased growth, defined as height and/or head circumference at least two standard deviations above the mean, associated with facial dysmorphism and intellectual disability occurring in patients with de novo heterozygous germline mutations in *DNMT3A*[1]. In the initial report, 13/152 patients with overgrowth had mutations in the *DNMT3A* gene, and all mutations were located within the functional domains of the protein. A subsequent publication assessing 55 patients[2] and numerous case reports[3–11] confirmed the original findings, and also described an additional array of mutations scattered throughout the functional domains of *DNMT3A*, including missense, frameshift, and nonsense mutations. For the DOS patients described to date, the most prevalent mutations alter amino acid position R882, which is also the most common site of *DNMT3A* mutations in patients with AML[12]. To date, 12/100 patients identified in the literature have mutations affecting R882[2–6,13]. More than 80% of individuals with DOS had overgrowth and a variable degree of intellectual disability. Obesity (weight of more than two standard deviations above the age and sex-adjusted mean) has been reported in two-thirds of patients. A characteristic facial appearance with heavy, horizontal, and low-set eyebrows with prominent and enlarged upper central incisors are frequent findings. Some patients have been reported to have joint hypermobility, hypotonia, kyphoscoliosis, and afebrile seizures[2].

Despite the emerging phenotypic and molecular characterization of DOS, limited data are available on the epigenetic consequences of the broad array of *DNMT3A* mutations seen in these patients. The precise consequences of most of the *DNMT3A* mutations are neither yet clear, nor are genotype: phenotype correlations for patients with this syndrome.

Somatic mutations in *DNMT3A* are the most common cause of clonal hematopoiesis[14–16] and are the most common initiating mutation in AML patients with a normal karyotype[17–19]. Although mutations occur throughout the *DNMT3A* gene in AML patients, more than half occur at amino acid R882 (e.g., R882H, R882C, R882S, R882P, etc.)[12]. *DNMT3A*[R882] mutations encode a dominant-negative protein that is thought to act via two mechanisms: first, R882 mutant proteins fail to homodimerize, reducing the activity of the enzyme by ~80%, and second, R882 mutant proteins preferentially interact with the wild-type (WT) protein, creating a catalytic sink that traps the WT protein is inactive heterodimers (dominant-negative effect)[20,21]. Previous studies have shown that the reduced methyltransferase activity of cells heterozygous for an R882 mutation is associated with a focal, canonical hypomethylation phenotype at specific regions within the genomes of AML cells[17,20,22]. Further, we previously showed that the morphologically normal peripheral blood cells of one 9-year-old DOS patient with a heterozygous germline *DNMT3A*[R882H] mutation had a focal hypomethylation phenotype that was similar to that of AML cells with this mutation[22]. These data strongly suggest the hypomethylation phenotype precedes (and may contribute to) the development of AML. To date, the global methylation phenotypes of the DOS patients with non-R882 mutations (defined by WGBS) have not yet been described and compared to R882 mutations, although array-based methylation studies have been reported[5,23]. Furthermore, the risk of developing AML with germline *DNMT3A* mutations (especially R882) is not yet clear[4].

In this work, we investigate the DNA methylation phenotypes of peripheral blood cells from 11 DOS patients with *DNMT3A* mutations, including three at R882, and eight with alternative mutations (including one with a heterozygous deletion of the *DNMT3A* gene). We also describe the phenotype of mice with a germline *Dnmt3a*[R878H] mutation (the murine homologue of *DNMT3A*[R882H]), which includes many features of the human syndrome, including obesity, overgrowth, as well as behavioral and movement deficits. Whole-genome bisulfite and single-cell RNA-sequencing studies reveal overlapping methylation and gene-expression phenotypes between mouse and human hematopoietic cells. Finally, we show that *Dnmt3a*[R878H] mice spontaneously develop B-cell and myeloid malignancies with a long latency, suggesting that patients with DOS should be prospectively monitored for the development of hematologic cancers.

## Results

**DNMT3A overgrowth syndrome (DOS) patients have focal hypomethylation in nonleukemic hematopoietic cells.** To determine whether DOS patients have the DNA methylation changes in the genomes of their hematopoietic cells, we obtained peripheral blood samples from 11 children and adults with DOS. Clinical features are summarized in Table 1. Patients ranged in age from 20 months to 36 years when samples were collected, and all exhibited hallmarks of DOS, including overgrowth, intellectual and developmental delays, behavioral disorders (including autism, anxiety, and panic disorder), hypotonia, and distinct facial features. One patient (UPN 894912) was diagnosed with AML (French Amrican British classification subtype M4) 4 years before sample banking and was in morphologic complete remission when the test sample from the peripheral blood was collected. Three patients (UPN 624400, 154605, and 894912) had R882 mutations, and the remaining eight had unique mutations occurring throughout the *DNMT3A* gene; six were missense mutations, one was a nonsense mutation (UPN 228211) (Fig. 1a), and one (UPN 518693) had a heterozygous 135 kb deletion that encompassed the entire *DNMT3A* gene (Chr2:25,228,254–25,363,376; hg38) (Supplementary Fig. 1a).

To assess the methylation of genomic DNA in the blood cells of these patients, we performed whole-genome bisulfite sequencing (WGBS), with a median ~18x coverage of the human genome. We compared the methylation levels of 11 DOS samples to the peripheral blood samples of 15 healthy donors aged 4–43 years (*DNMT3A*[+/+]; eight male, seven female). We subcategorized the DOS patients into two groups: *DNMT3A*[R882] (n = 3; one male, two female) and *DNMT3A*[non-R882] (n = 8; five male, three female) for subsequent analysis, since each non-R882 patient had a unique mutation. There was no significant difference in age between control vs. *DNMT3A*[R882] (p = 0.157) or control vs. *DNMT3A*[non-R882] (p = 0.824) groups (two-sample t test). Globally, the methylation levels across the genome were subtly decreased in *DNMT3A*[R882] and *DNMT3A*[non-R882] patients (Fig. 1c). Utilizing established methods[24], we identified differentially methylated regions (DMRs) between *DNMT3A*[+/+] (n = 15) vs. *DNMT3A*[R882] samples (n = 3; Supplementary Data 1), and *DNMT3A*[+/+] vs. *DNMT3A*[non-R882] samples (n = 8; Supplementary Data 2). DMRs were defined as having ≥10 CpGs, a mean methylation difference of ≥0.2, a false discovery rate (FDR) of ≤0.05, and a standard deviation (SD) of ≤0.1 among the test samples. To ensure that the 2,209 DMRs called in the R882 samples were independent of age and/or sex effects on CpG methylation, we used linear regression to test for the effect of genotype on methylation level while adjusting for sex and log (age). All of the DMRs remained significant (at FDR < 0.05) in this regression analysis. At DMRs, the mean methylation values of *DNMT3A*[R882] samples were lower than the *DNMT3A*[non-R882] samples at all annotated regions of the genome examined,

**Table 1 Patient characteristics.**

| UPN | Age at sample collection (yrs) | DNMT3A mut | Gender | Growth phenotype | Behavioral/intellectual | History of cancer |
|---|---|---|---|---|---|---|
| 624400 | 9 | R882H | M | Overgrowth | Global developmental delay, IQ 50 | No |
| 154605 | 1.7 | R882H | F | Overgrowth | Global developmental delay, limited vocab | No |
| 894912 | 16, in AML remission | R882C | F | Overgrowth | Global developmental delay | AML, FAB M4, age 12 |
| 411168 | 17 | C583Y | M | Overgrowth | Autism, IQ 40 | No |
| 511909 | 5 | R301W | M | Overgrowth | Autism, developmental delay | No |
| 228211 | 34 | F414fsTer7 | M | Overgrowth | ADHD, panic disorder, intellectual delay | Dermatofibrosarcoma age 20 (excision) |
| 295041 | 36 | R736H | F | Overgrowth | OCD, Depression, anxiety | No |
| 930075 | 16 | R688H | F | Overgrowth | Global developmental delay | No |
| 723972 | 3 | Y660H | M | Overgrowth | Global developmental delay | No |
| 518693 | 5 | 135kb del | M | Overgrowth | ADHD, autism spectrum disorder | No |
| 786396 | 16 | I310N | F | Overgrowth | ADD, Autism, IQ approx 70, OCD, anxiety | No |

including gene bodies and promoters ($p \leq 0.0001$, Fig. 1b, c). However, the average width of DMRs (in base pairs) in the $DNMT3A^{R882}$ samples ($662.7 \pm 441.6$) was not statistically different from those in the $DNMT3A^{non-R882}$ ($614.5 \pm 366.8$, $p = 0.0587$; Fig. 1d). Interestingly, the fraction of gene bodies containing at least one DMR (1426/18951, 7.52%) was significantly greater than that observed in other annotated regions of the genome (0.06–2.4%; ***$p \leq 0.0001$; Supplementary Fig. 1b).

The relative severity of the methylation phenotype in the $DNMT3A^{R882}$ samples compared to $DNMT3A^{non-R882}$ (as well as the canonicality of the phenotype among samples within a category) is shown in a heatmap representation of the 2209 DMRs identified by comparing the healthy $DNMT3A^{+/+}$ donors vs. $DNMT3A^{R882}$ samples (Fig. 1e). Passively plotting the methylation values of the non-R882 samples revealed the attenuated methylation difference in the same regions in those samples. In contrast, the direct comparison of $DNMT3A^{non-R882}$ samples to healthy donors revealed only 332 DMRs (Fig. 1f). In both comparisons, all DMRs identified were hypomethylated, and 215 DMRs (65% of $DNMT3A^{non-R882}$ and 10% of $DNMT3A^{R882}$ DMRs) overlapped between $DNMT3A^{R882}$ and $DNMT3A^{non-R882}$ samples (where overlap required at least 1 bp of shared sequence). An example of a region within the *HOXB* cluster with DMRs (highlighted by black boxes) is shown in Fig. 1g, along with two striking DMRs in exons four and five of the *RASIP1* gene, that are specific for the R882 samples (Fig. 1g). The DMRs present in these regions are more pronounced in the R882 mutant samples, suggesting that R882 mutations cause a greater reduction in DNA methyltransferase activity than the non-R882 mutations.

**$DNMT3A^{R882}$ alters the transcriptional signatures of hematopoietic cells**. Utilizing the 10x Genomics Chromium platform[25], we performed single-cell RNA-sequencing (scRNA-seq) on fresh peripheral blood samples from UPN 624400 (at age 14) and his unaffected male sibling (at age 17). The two samples showed a high level of similarity (Supplementary Fig. 2a). Graph-based clustering identified 12 clusters that were functionally categorized with ToppGene[26] by inputting the top 50 identifying genes for each cluster (Fig. 2a). These populations were then validated by the assessment of well-established gene markers enriched in different subsets (Supplementary Fig. 2b). The cell type distribution was different between the $DNMT3A^{R882H}$ patient and his sibling control ($p < 0.0001$, chi-squared), and subtle differences in several individual populations (relative to total cells) were observed (Fig. 2b). The fraction of CD4 + naïve T-cells (cluster 5) was reduced (13.94–4.77%), and the NK-cell (cluster 8) and NKT (cluster 10) fractions were increased (3.09–15.88% and 3.80–7.01%, respectively, Fig. 2b). The reduction in T-cells and increase in NK-cells was orthogonally validated by 15-color flow cytometry of the same samples (Supplementary Fig. 2c). We identified differentially expressed genes in each cluster in the $DNMT3A^{R882H}$ vs. sibling control samples with a $p$-value and expression cutoff of 0.05 and log2 ratio of ±1, respectively (Supplementary Data 3). Out of 12 clusters, nine had differentially expressed genes (DEGs; Fig. 2c); however, the total numbers of DEGs were relatively small, ranging from five (cluster 11) to 72 (cluster 1). Interestingly, 50 genes were identified as dysregulated in two or more clusters, and their dysregulation was canonical across cell types, suggesting they were dysregulated by mechanisms not specific to lineage or cell type (Fig. 2d). For example, the *RASIP1* gene (which contains two DMRs; Fig. 1g) was upregulated in 7/9 clusters with detectable DEGs (Fig. 2e). This was due to an increase in the number of expressing cells, rather than an increase in the expression level of *RASIP1* per cell, since the control sample had a single cell with detectable *RASIP1*

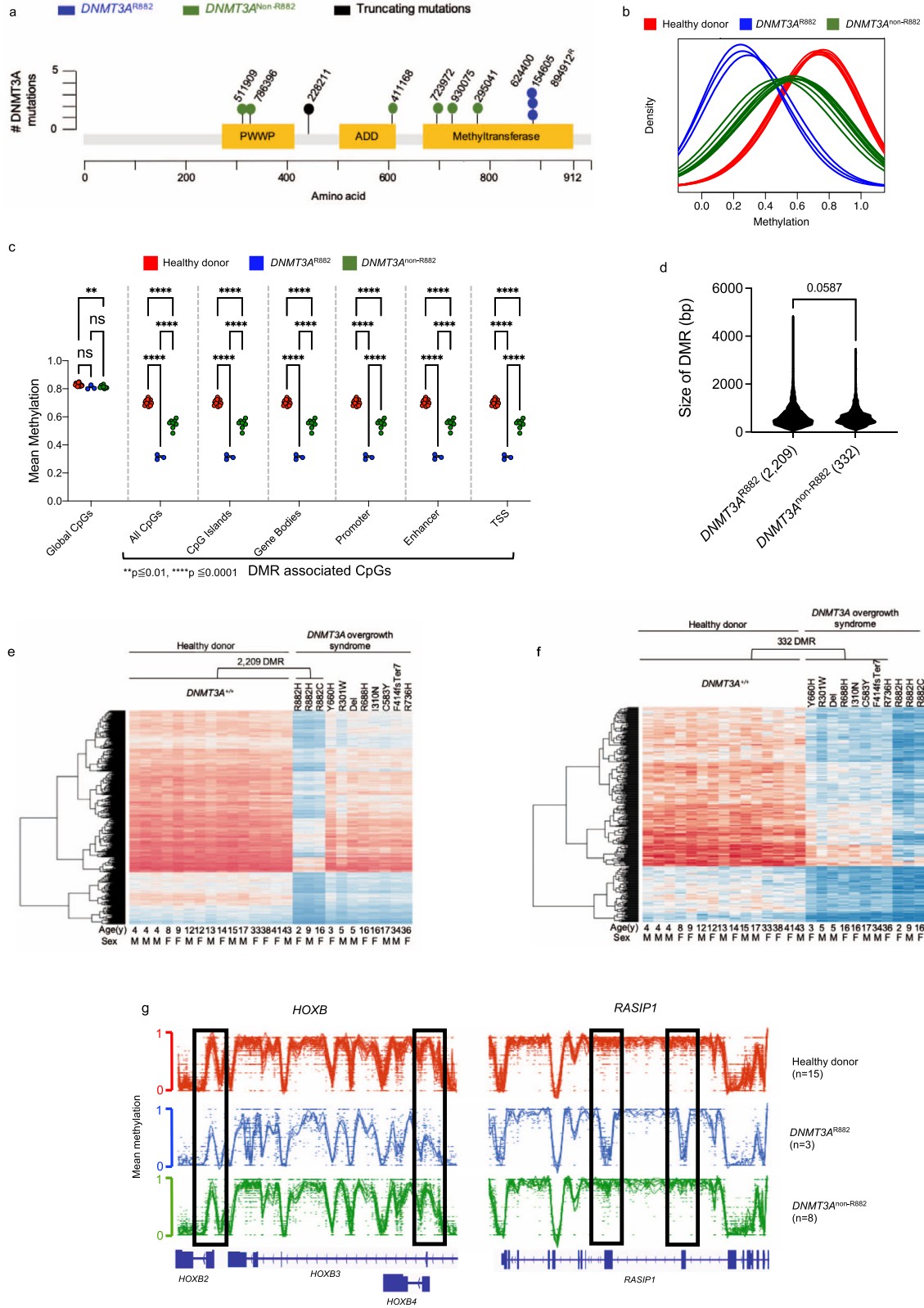

expression, while in the $DNMT3A^{R882H}$ sample, 15% of cells had detectable reads. In contrast, $HOXB$ genes, also associated with DMRs, were downregulated across multiple cell types. $HOXB2$ was downregulated due to a decrease in the number of expressing cells, and significantly decreased mean reads per cell (Fig. 2e).

However, there was no direct correlation between DMR location and expression of the $HOXB$ genes, suggesting there may be additional local or long-range regulatory elements within the cluster that influence the expression of $HOXB$ genes[27]. Gene Ontology terms for DEGs by cluster identified numerous

**Fig. 1 Peripheral blood cells of DOS patients with *DNMT3A*[R882] mutations have a more severe methylation phenotype than non-R882 mutations. a** Distribution of germline *DNMT3A* mutations identified in DOS patients from this study. **b** Density plot of methylation values from whole-genome bisulfite sequencing (WGBS) for CpGs within DMRs for each peripheral blood sample from healthy donors (red; *n* = 15), *DNMT3A*[R882] (blue, *n* = 3), and *DNMT3A*[non-R882] (green, *n* = 8) patients. **c** Mean methylation values for both global CpGs and DMR-associated CpGs in specific, annotated regions of the genome. Hypothesis testing was performed via two-way repeated-measures ANOVA with Tukey's multiple comparison test within each genomic region. (ns = not significant, **$p \leq 0.01$, ****$p \leq 0.001$). **d** Mean Size (in bp) of DMRs identified in *DNMT3A*[R882] (*n* = 2,209) and *DNMT3A*[non-R882] (*n* = 332) peripheral blood samples (*P* = 0.0587, two-tailed *t* test). **e** Heatmap showing the mean methylation values for the 2209 DMRs defined in *b* for each individual healthy donor and *DNMT3A*[R882] sample. The values for the same DMRs were also plotted passively for *DNMT3A*[non-R882] samples (age and sex shown below). **f** Heatmap showing the mean CpG methylation values for the 332 DMRs defined by comparing the healthy donors and *DNMT3A*[non-R882] samples. Values for the same DMRs were plotted passively for *DNMT3A*[R882] samples (age and sex shown below). **g** Examples of DMRs within the *HOXB* cluster and *RASIP1* gene. Healthy donors are shown in red, *DNMT3A*[R882] cases in blue, and *DNMT3A*[non-R882] cases in green. Gene tracks are shown below, and DMRs are designated in boxes. DMR differentially methylated region, bp base pairs, TSS transcriptional start site.

biological processes, including the terms related to T-cell activation and proliferation (IL-4 and IFNG signaling), and vascular function (angiogenesis and endothelial cell migration) (Supplementary Fig. 2d).

Across all 12 clusters, there were 242 differentially expressed genes in total; only 38 of these were within 10 kb of a DMR (Supplementary Fig. 2e). This suggests that for the majority of DEGs, there is no direct correlation to a local DMR. Using bulk RNA-sequencing, we confirmed the presence of DEGs in this *DNMT3A*[R882H] patient, as well as one additional *DNMT3A*[R882H] patient (aged 1.7 years), including validation of *RASIP1* and *HOXB2* dysregulation (Supplementary Fig. 2f, g). While bulk RNA-seq was able to identify additional DEGs (in part to increased sequencing depth), the lack of a global correlation between differential gene expression and DMRs was recapitulated, even across specific annotated regions of the genome (Supplementary Fig. 2h and Supplementary Data 4).

**Mice with germline *Dnmt3a*[R878H/+] exhibit overgrowth and obesity.** To achieve germline expression of the *Dnmt3a*[R878H/+] allele from its endogenous locus, we utilized the model established by Guryanova et al. a minigene combining exons 23 and 24 carrying the point mutation encoding R878H was inserted in the place of the endogenous *Dnmt3a*[+/+] exon 23 downstream of a lox-stop-lox cassette[28]. Heterozygous *Dnmt3a*[R878H/+] mice were crossed with B6.C-Tg(CMV-cre)1Cgn/J deleter mice. The floxed founder line mice with *Dnmt3a*[R878H/+] and CMV-cre were then backcrossed to C57Bl/6 J mice to transmit the mutant allele through the germline, and to select for mice without CMV-Cre. Heterozygous floxed *Dnmt3a*[R878H/+] mice were born at the expected ratios (for over 30 genotyped litters, the ratio of *Dnmt3a*[R878H/+] to *Dnmt3a*[+/+] was 1.105) and were viable, surviving to 2+ years. Expression of the R878H allele after floxing was virtually identical to that of the WT allele[28]. *Dnmt3a*[R878H/R878H] mice were severely runted at birth, and did not survive past 1 week of age. Female heterozygous *Dnmt3a*[R878H/+] mice had a significant incidence of dystocia during pregnancy, and were therefore not utilized to produce experimental mice. All experimental mice were generated by crossing male heterozygous *Dnmt3a*[R878H/+] germline mice (CMV-Cre negative) with C57Bl/6 J female mice. Germline *Dnmt3a*[R878H/+] mice had normal weight and size at birth, and no obvious developmental defects.

Many DOS patients have increased height and obesity[2]. We therefore tracked weight for heterozygous mutant *Dnmt3a*[R878H/+] mice (*n* = 120; 59 females, 61 male) and *Dnmt3a*[+/+] littermate controls (*n* = 90; 48 females, 42 male) from 21 to 600 days of age. Before 100 days of age, there was no difference in weights of the two cohorts (*p* = 0.5781 for genotype*time). However, with aging, the weights of the R878H mice diverged from littermate controls, reaching a mean of 37.73 and 31.2 g at 380 days of age, respectively (*p* ≤ 0.0001 for genotype*time; Fig. 3a). Visual

inspection revealed a clear size difference for the mutant mice at 1 year of age (Fig. 3b). CT scans performed on four pairs of 210-day old mice showed significantly longer femur lengths (but not humerus lengths), which represents a surrogate for increased height (Fig. 3c–e). We also measured craniofacial landmarks to determine whether *Dnmt3a*[R878H/+] mice had macrocephaly, a phenotype often observed in DOS children[1,2]. Although statistical analyzes of skull measurements did not show a difference in head circumference (length of and width of the neurocranial bones), some features of the skulls of *Dnmt3a*[R878H/+] mice are slightly larger than their WT littermates, including the mandible and localized structures in the cranium (Fig. 3f). The MRI measurements of body composition revealed a significant, age-dependent increase in body fat (Fig. 3g), but not lean mass (Fig. 3h) paralleling the obesity phenotype observed in DOS patients. Obesity was not associated with an increase in food consumption; *Dnmt3a*[R878H/+] mice ate significantly less chow than WT-aged matched controls (mean = 3.51 ± 0.07 vs. 4.33 ± 0.17 grams/mouse/day, *p* = 0.0013; Fig. 3i). In addition, metabolic cage analysis confirmed that *Dnmt3a*[R878H/+] mice ate less, a finding that was associated with slightly (albeit not significantly) reduced movement and overall speed of movement (Supplementary Fig. 3a–c). The metabolic cage analysis also revealed that *Dnmt3a*[R878H/+] mice under 6 months of age did not have significant differences in $O_2$ consumption or $CO_2$ expiration (although both trended toward reduced levels; Supplementary Fig. 3d, e). In mice over 6 months of age, there was a decrease (although not significant) in $O_2$ consumption and a significant decrease in $CO_2$ production (*p* = 0.0313). Together, this suggests that there is a slight decrease in the respiratory quotient (RQ) in *Dnmt3a*[R878H/+] mice (Supplementary Fig. 3f); although not significant, this change may indicate a subtle reduction in basal metabolic rate in these mice, contributing, at least in part, to obesity.

The obesity phenotype was exacerbated by feeding a high-fat diet, which did not cause a similar weight gain in *Dnmt3a*[+/+] littermates (Fig. 3j). Finally, we fasted mice for 6 h, and then assayed plasma for common analytes relevant to increased adiposity. The plasma levels of leptin, triglycerides, cholesterol, glucose, and free fatty acids were not significantly different between *Dnmt3a*[R878H/+] and *Dnmt3a*[+/+] control mice (Supplementary Fig. 3g–o).

**Mice with germline *Dnmt3a*[R878H/+] exhibit behavioral abnormalities.** To assess behavioral and neurological phenotypes in the *Dnmt3a*[R878H/+] mice, we carried out a battery of tests on littermate and age-matched cohorts of WT and R878H mice (all mice were 100–200 days old at the time of testing, Fig. 4 and Supplementary Fig. 4); and the measurement goals are detailed in the "Methods" section. A number of results were significantly altered in *Dnmt3a*[R878H/+] mice, including reduced total ambulations and rearing events in 1 h open field tests, reduced pole climb down and inverted screen (60 and 90°) climb-up times, increased time spent freezing in contextual, conditional and cued fear testing, differential

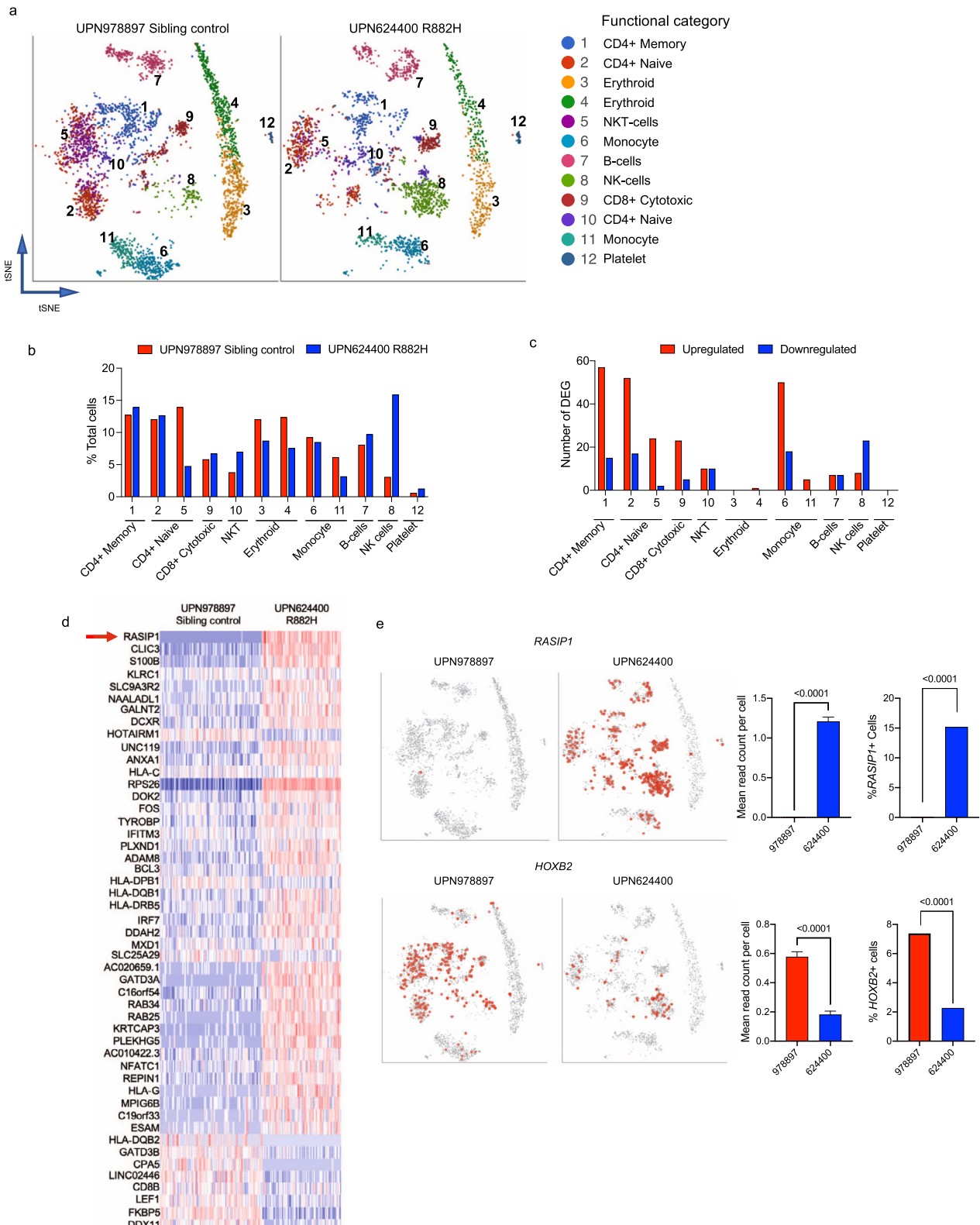

foot-shock response, and reduced marble-burying (Fig. 4a–k, respectively). Outcomes that were not statistically different in R878H mice included the sensorimotor battery, such as balance (ledge test and platform test), walking initiation, grip strength (inverted screen test) and motor coordination (rotarod), as well as the intellectual disability, memory and anxiety tests, including the elevated plus maze, Morris water maze test, probe trial (Supplementary Fig. 4).

These data indicate that $Dnmt3a^{R878H/+}$ mice display reductions in volitional movement with an absence of frank changes to exploratory behavior, suggestive of complex emotionality in these mice. Overall, the behavioral analyses demonstrated that germline $Dnmt3a^{R878H/+}$ mice have a predominant phenotype of volitional movement deficits, accompanied by complex emotionality, and subtle cognitive alterations.

**Fig. 2 Peripheral blood cells of a DOS patient with an *DNMT3A*R882 mutation have differentially expressed genes. a** tSNE projection of scRNA-seq data from peripheral blood derived from one DOS patient with an R882 mutation (right; UPN 624400, age 14 at collection) and his matched *DNMT3A*+/+ sibling control sample (left; UPN 978897, age 17 at collection). Graph-based clustering identified 12 distinct clusters that were functionally categorized by gene-expression analysis utilizing Toppfun. **b** The percentage contribution of each graph-based cluster associated with scRNA-seq data shown in Panel *a*. **c** Numbers of upregulated (red bars) and downregulated (blue bars) genes in each graph-based cluster. **d** The heatmap of 50 differentially expressed genes identified as dysregulated in more than one graph-based cluster. **e** Examples of two differentially expressed genes (*HOXB2* and *RASIP1*) identified in more than one graph-based cluster (highlighted in Panel *d* with red arrows) and associated with a differentially methylated region; tSNE projections, mean read counts per cell + SEM, and percentage of cells expressing each gene are shown (P-values by *t* test or Fisher's exact test for ratio's are indicated, n = 1 biologically independent samples per genotype). DEG differentially expressed genes.

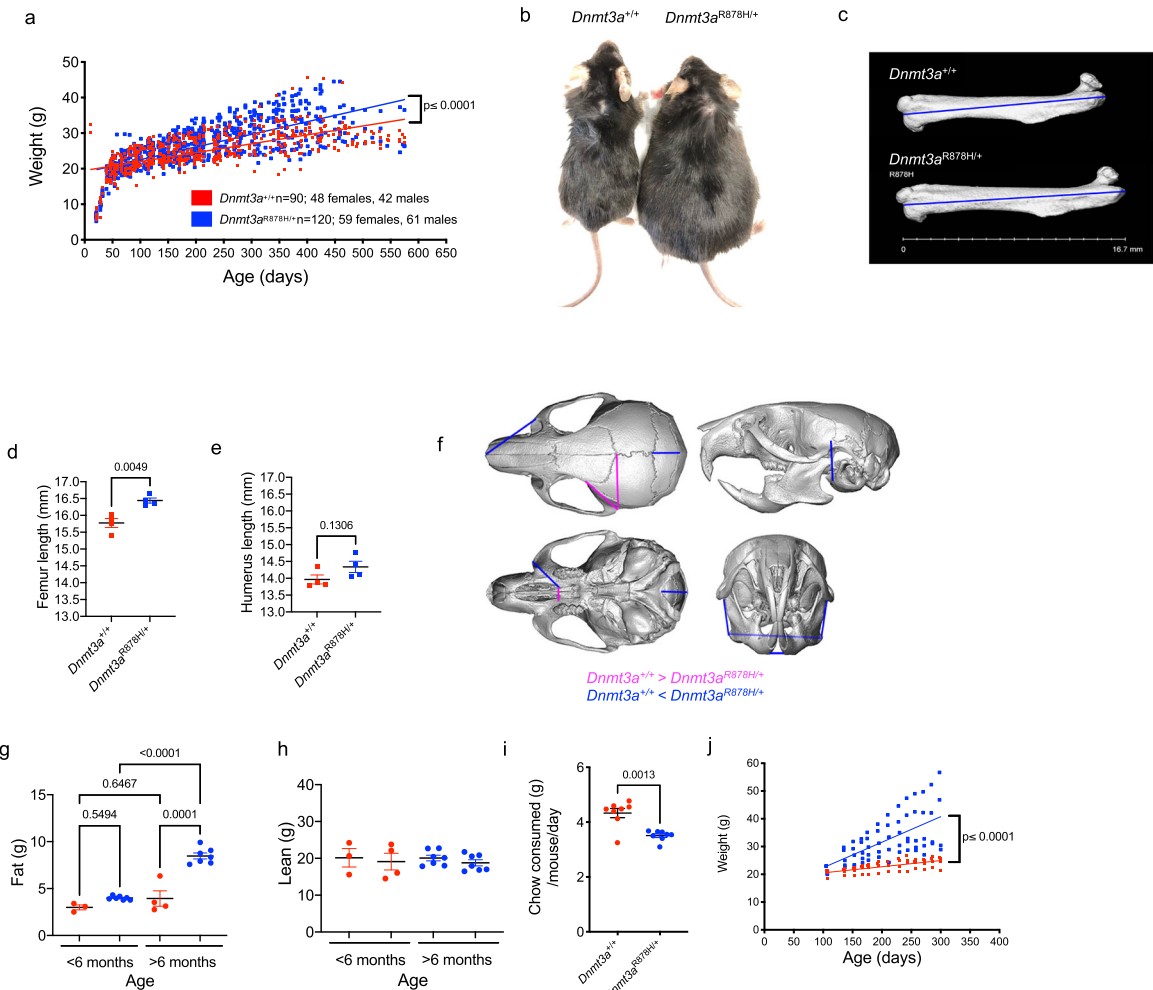

**Fig. 3 Germline *Dnmt3a*R878H/+ mice exhibit overgrowth and obesity. a** Weight in grams of *Dnmt3a*+/+ control mice (red, *n* = 90) and *Dnmt3a*R878H/+ mice (blue, *n* = 120) from weaning (day 21) to 575 days of age. Linear regression analysis was used to determine differences between genotypes (*p* ≤ 0.0001). **b** Representative image of age and gender-matched littermate *Dnmt3a*+/+ (30.9 g) vs. *Dnmt3a*R878H/+ (59.45 g) mice highlighting a typical size difference at 1 year of age. **c** Representative CT images of femurs from whole mouse imaging. Measurements of femur length (mm) in *Dnmt3a*+/+ and *Dnmt3a*R878H/+ age-matched controls are shown (*n* = 4 each genotype). **d** Quantification of femur lengths (mm) measured as shown in *b* (*n* = 4 biologically independent samples/genotype, *P* = 0.0049 by two-tailed *t* test). **e** Quantification of humerus lengths (mm) in the same mice, measured by CT scan (*n* = 4 biologically independent samples/genotype, p-value by two-tailed *t* test). **f** Reconstructions of cranium and mandible from CT scans of *Dnmt3a*+/+ and *Dnmt3a*R878H/+ mice showing landmarks of differences in size and angles. **g** The MRI quantification of body fat composition of age-matched *Dnmt3a*+/+ (red, *n* = 4) and *Dnmt3a*R878H/+ (blue, *n* = 7) pairs separated by age (less or greater than 6 months), to highlight age-dependent increases in body fat (*P*-values by two-way ANOVA with Tukey's multiple comparisons test are indicated). **h** The MRI quantification of lean body mass composition of age-matched *Dnmt3a*+/+ (red, *n* = 4) and *Dnmt3a*R878H/+ (blue, *n* = 7) pairs separated by age (less or greater than 6 months). **i** Quantification of chow consumed (calculated as grams of chow per mouse per day) for age-matched *Dnmt3a*+/+ (red, *n* = 8) and *Dnmt3a*R878H/+ (blue, *n* = 8) mice (*P* = 0.0013 by two-tailed *t* test). **j** Weight tracking in grams of *Dnmt3a*+/+ control mice (red, *n* = 4) and *Dnmt3a*R878H/+ mice (blue, *n* = 6) fed a high-fat diet from 100 to 300 days of age. Linear regression analysis was used to determine differences between genotypes (*p* ≤ 0.0001). For d, e, g-i error bars show mean ± SEM. g grams, mm millimeter.

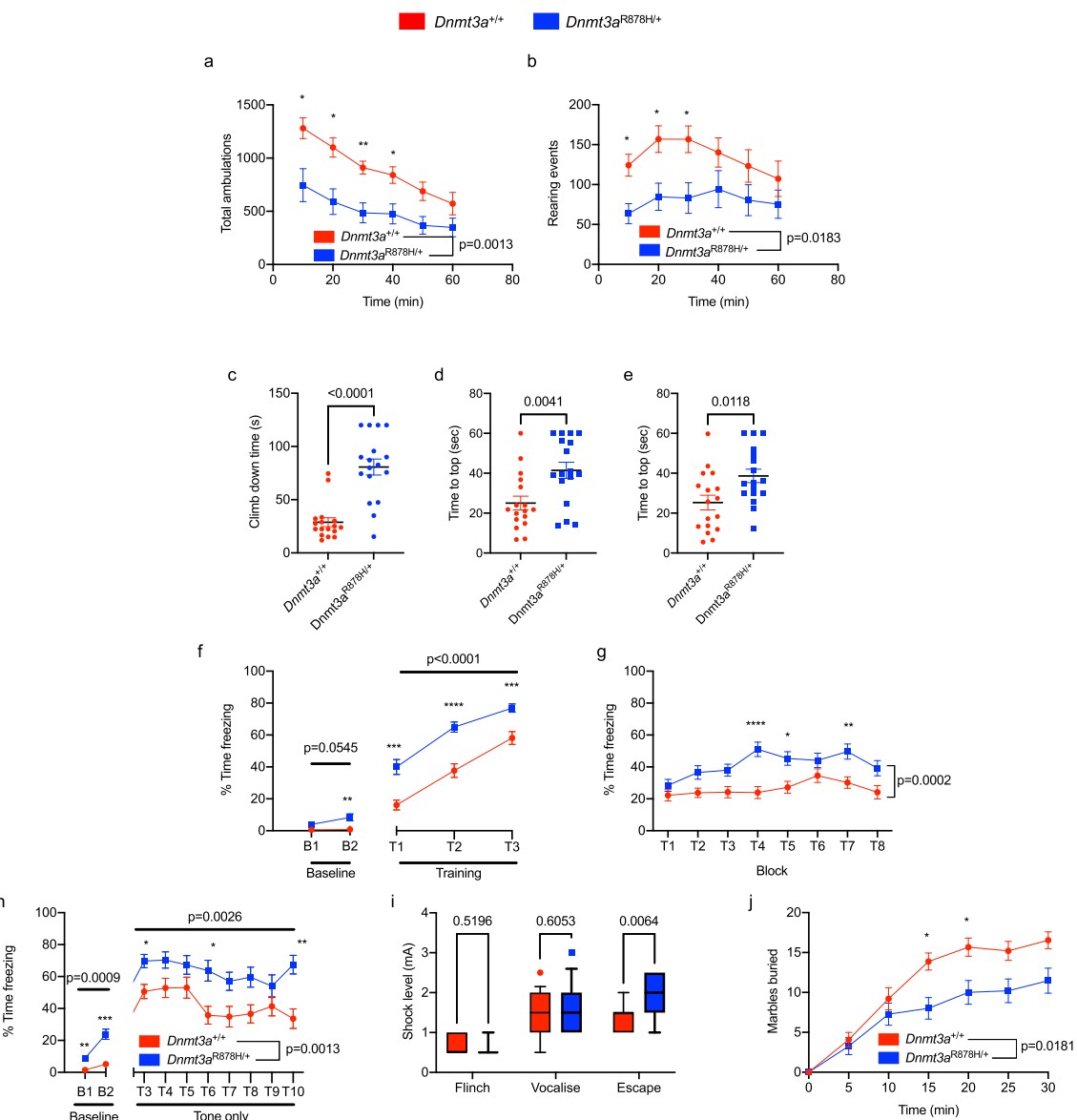

**Fig. 4 Germline *Dnmt3a*^R878H/+ mice exhibit behavioral alterations. a** Total ambulations during 1-hour open-field testing, split into 10-minute intervals ($p = 0.0013$ effect by genotype, $F_{(1,32)} = 12.5$, $n = 17,17$; two-way repeated-measures ANOVA with Šídák's multiple comparison test). **b** The number of rearing events during 1-hour open-field testing, split into 10-minute intervals ($p = 0.0183$ effect by genotype, $F_{(1,32)} = 6.183$, $n = 17,17$; two-way repeated-measures ANOVA with Šídák's multiple comparison test). *Dnmt3a*^R878H/+ mice take significantly longer to **c** climb down a pole ($p \leq 0.0001$, $n = 17,17$; unpaired *t* test), reach the top of a **d** 60° inclined screen ($p = 0.0041$, $n = 17,17$; unpaired *t* test) and **e** 90° inclined screen ($p = 0.0118$, $n = 17,17$; unpaired *t* test). Panels **f**–**h** The percentage of time freezing in **f** conditioned fear training (cue: $p \leq 0.0001$ genotype effect, $F_{(1,31)} = 28.02$, $n = 16,17$; two-way repeated-measures ANOVA with Šídák's multiple comparison test), **g** contextual fear trials ($p = 0.0002$ genotype effect, $F_{(1,31)} = 18.53$, $n = 16,17$; two-way repeated-measures ANOVA with Šídák's multiple comparison test) and **h** cued fear trials (baseline: $p = 0.0009$ genotype effect, $F_{(1,31)} = 31.02$, $n = 16,17$; cue: $p = 0.0026$ genotype effect $F_{(1,31)} = 10.76$, $n = 16,17$; two-way repeated-measures ANOVA with Šídák's multiple comparison test). **i** Minimum shock needed to exhibit a behavioral response in mice during conditioned fear test (escape: $p = 0.0064$ genotype effect $F_{(1,31)} = 5.094$, $n = 16,17$; two-way repeated-measures ANOVA with Šídák's multiple comparison test). The box extends from 25th to 75th percentile, the line indicates median and whiskers show 10th to 90th percentile. **j** Marbles buried in 30 min (5 min bins, $p = 0.0181$, genotype effect $F_{(1,28)} = 6.307$, $n = 15,15$; two-way repeated-measures ANOVA with Šídák's multiple comparison test). *$p \leq 0.05$, **$p \leq 0.01$, ***$p \leq 0.001$, ****$p \leq 0.0001$. The line and bar graphs indicate the mean and SEM for each value tested. min minutes, s seconds, mA milliamps.

**Germline *Dnmt3a*^R878H/+ mice have a focal hypomethylation phenotype in nonleukemic hematopoietic cells.** To understand the methylation phenotypes of bone marrow cells derived from unmanipulated, littermate-matched *Dnmt3a*^+/+ ($n = 10$; 2–52 weeks of age, five male, five female) or *Dnmt3a*^R878H/+ mice ($n = 6$; 8–38 weeks of age, three male, three female), we performed WGBS. There was no significant difference in age between control and *Dnmt3a*^R878H/+ groups ($p = 0.779$; two-

sample *t* test). We also included the bone marrow cells from unmanipulated germline *Dnmt3a*^-/- mice ($n = 4$; 2 weeks of age, two male, two female) and *Dnmt3a*^+/- mice ($n = 4$; 12–52 weeks of age, three male, one female) as comparators to calibrate the methylation phenotypes of *Dnmt3a*^R878H/+ relative to deficiency or haploinsufficiency. The DNA methylation phenotypes of *Dnmt3a*^-/- [29] and *Dnmt3a*^+/- [30] models have been described in the literature[31–33], and will not be further discussed here.

Our WGBS sequence coverage (median of 18x) assessed 98% of individual CpGs in the mouse genome. The methylation values had Pearson's correlation $r > 0.8$ within samples for each genotype, highlighting the reproducibility between biological replicates, despite the range in ages for the mice used in the study. Globally, the mean methylation differences among all CpG sites in the $Dnmt3a^{R878H/+}$ bone marrow samples were not significantly different from the $Dnmt3a^{+/+}$ samples (Fig. 5a). In contrast, $Dnmt3a^{-/-}$ samples had significantly reduced mean global methylation across all CpGs. We next defined DMRs (using the same parameters used for the human samples) for $Dnmt3a^{+/+}$ vs. $Dnmt3a^{R878H/+}$ samples (#DMRs = 2172, Supplementary Data 5), $Dnmt3a^{+/+}$ vs. $Dnmt3a^{-/-}$ samples (#DMRs = 20161, Supplementary Data 6), and $Dnmt3a^{+/+}$ vs. $Dnmt3a^{+/-}$ samples (#DMRs = 8, Supplementary Data 7) and found that the $Dnmt3a^{R878H/+}$ phenotype was intermediate between the $Dnmt3a^{-/-}$ and $Dnmt3a^{+/+}$ mice (Fig. 5b). To ensure that the 2172 DMRs called in the R878H samples were independent of age and sex effects on CpG methylation, we used linear regression to test for the effect of genotype on methylation level, while adjusting for sex and log(age). All of the DMRs remained significant (at FDR < 0.05) in this regression analysis. Comparison of the 2172 $Dnmt3a^{R878H/+}$-specific DMRs with the $Dnmt3a^{-/-}$ samples revealed virtually complete overlap, but the degree of methylation reduction in the $Dnmt3a^{R878H/+}$ samples was uniformly less severe. Within those 2172 DMRs, $Dnmt3a^{R878H/+}$ samples had a mean CpG methylation value of 50.10%, $Dnmt3a^{-/-}$ DMRs had a mean methylation value of 33.35%, and $Dnmt3a^{+/-}$ DMRs had a mean methylation value of 85.5% compared to that of $Dnmt3a^{+/+}$ controls (defined as having 100% methylation within the same DMRs). This trend was observed across all annotated regions of the genome and suggests that the $Dnmt3a^{R878H}$ protein behaves as a dominant-negative at all functional regions of the genome (Fig. 5a). In addition, the mean size of DMRs was $890.1 +/- 552$ bp for $Dnmt3a^{-/-}$ samples, $751.1 +/- 474.2$ bp for the $Dnmt3a^{R878H/+}$ samples, and $610.3 +/- 295.7$ bp for the $Dnmt3a^{+/-}$ samples (Fig. 5c).

Of 22,026 gene bodies annotated in the mouse genome, 1375 (6.24%) contained at least one DMR (Supplementary Fig. 1c), a significant enrichment compared to other annotated regions in the genome ($p \leq 0.0001$). In human peripheral blood, 7.52% of gene bodies contained a DMR (Supplementary Fig. 1b). The fractions of other annotated regions associated with DMRs were also similar between human and mouse samples, suggesting there is an overlap of the functional consequences of human R882 and mouse R878H mutations.

The focal and canonical nature of the 2172 and 20,161 DMRs identified in $Dnmt3a^{R878H/+}$ and $Dnmt3a^{-/-}$ bone marrow samples, respectively, are highlighted in Fig. 5d, e. The methylation pattern across all samples within a genotype was highly reproducible. From these heatmaps, it was also clear that essentially all DMRs in $Dnmt3a^{R878H/+}$ samples are likewise hypomethylated in the $Dnmt3a^{-/-}$ samples. An intersection analysis revealed that 81.4% of $Dnmt3a^{R878H/+}$ DMRs were also detected in $Dnmt3a^{-/-}$ samples; in contrast, the $Dnmt3a^{+/-}$ samples had very few DMRs and most closely resembled $Dnmt3a^{+/+}$ samples (Fig. 5d). The dramatic methylation loss at DMRs in $Dnmt3a^{-/-}$ bone marrow was less severe in the $Dnmt3a^{R878H/+}$ samples (Fig. 5e). Examples of the focality and canonicality of methylation changes at specific, homologous loci are demonstrated for the $Hoxb$ cluster and the $Rasip1$ gene (Fig. 5f), where similarly located DMRs were detected in human R882 samples. To compare DMRs across species we used the UCSC lift-over tool (http://genome.ucsc.edu) to translate the human R882 DMR coordinates to the mouse genome, and found that 101 human DMRs (4.6%) directly corresponded to a mouse R878H DMR and 1713 (77.5%) were within 10 kb of a mouse DMR. Conversely, when we lifted the mouse coordinates over to the human genome, 95 (4.37%) mouse DMRs intersected directly with a human DMR, and 1889 (87%) were within 10 kb of a human R882 DMR.

**Germline $Dnmt3a^{R878H/+}$ mice have differentially expressed genes in specific hematopoietic cell types.** We performed scRNA-seq on whole bone marrow cells from two pairs of $Dnmt3a^{+/+}$ and $Dnmt3a^{R878H/+}$ mice using the 10x Genomics Chromium platform[30]. One littermate-matched pair was evaluated at 1 month of age, and the other at 9 months, and the paired samples showed remarkable similarity by tSNE (Supplementary Fig. 5a). After processing aligned data with Partek Flow software, we performed graph-based clustering and ToppGene analysis to identify functional cell types based on the top 50 defining genes for each cluster (Fig. 6a, b). Inferred lineages for each cluster were verified using a $k$-nearest neighbor algorithm trained on the Haemopedia Database[34] (Supplementary Fig. 5b), and all lineages were present in both genotypes (Fig. 6a, b, Supplementary Fig. 5a–c). There were subtle, yet significant, differences in pre-B-cells, monocytes, macrophages, and MPPs in the $Dnmt3a^{R878H/+}$ bone marrow relative to controls (by Fisher's exact tests for ratios) suggesting that the expression of germline R878H does not lead to large disturbances in normal hematopoietic populations. Graph-based clustering identified one population that was expanded in the 9-month $Dnmt3a^{R878H/+}$ sample that did not fit into a single lineage assignment because it had both B-cell and myeloid gene-expression signatures (cluster 14; mixed lineage).

We next assessed how many differentially expressed genes overlapped between the 1- and 9-month samples (6047 and 5573 unique DEGs, respectively across 24 clusters), and found 2,775 DEGs common to both ages (Supplementary Data 8). A comparison of differentially expressed genes identified in scRNA-seq data from $DNMT3A^{R882}$ human peripheral blood (242 total unique DEG across 12 clusters, Fig. 2) and mouse $Dnmt3a^{R878H/+}$ bone marrow (8,845 total genes across 1- and 9-month timepoints) showed concordant dysregulation for 121 genes (50% of human genes) based on gene name alone. We show examples of conserved dysregulation in the $Hoxb$ gene cluster ($Hoxb4$; Fig. 6c) and for $Rasip1$ (Fig. 6d).

To further validate the dysregulation of $Hoxb$ gene expression in $Dnmt3a^{R878H}$ cells, we evaluated its expression in a previously published $Dnmt3a^{-/-}$ scRNA-seq dataset that also utilized a doxycycline-inducible, WT $DNMT3A$ transgene to restore DNA methylation[29]. Inducing DNMT3A activity by feeding mice dox chow results in time-dependent remethylation at DMRs, with partial remethylation of the $Hoxb$ locus occurring at approximately 24 weeks (Supplementary Fig. 6a)[29]. In $Dnmt3a^{-/-}$ hematopoietic cells from these mice, $Hoxb4$ expression was likewise decreased, predominantly in myeloid lineage cells (PMNs; Supplementary Fig. 6b, monocytes; Supplementary Fig. 6c, and GMPs; Supplementary Fig. 6d); restoring $DNMT3A$ expression with doxycycline in vivo led to a time-dependent increase in mean $Hoxb4$ expression per cell, and an increase in the fraction of myeloid cells expressing $Hoxb4$ (Supplementary Fig. 6b–d). These data suggest that the DNA methylation status of the $Hoxb$ cluster may directly influence the expression of $Hoxb$ genes.

**Hematopoietic phenotypes and spontaneous leukemias in germline $Dnmt3a^{R878H/+}$ mice.** The sizes of cell populations defined in the scRNA-seq data suggest that steady-state hematopoiesis is relatively unperturbed in $Dnmt3a^{R878H/+}$ mice. Using 21

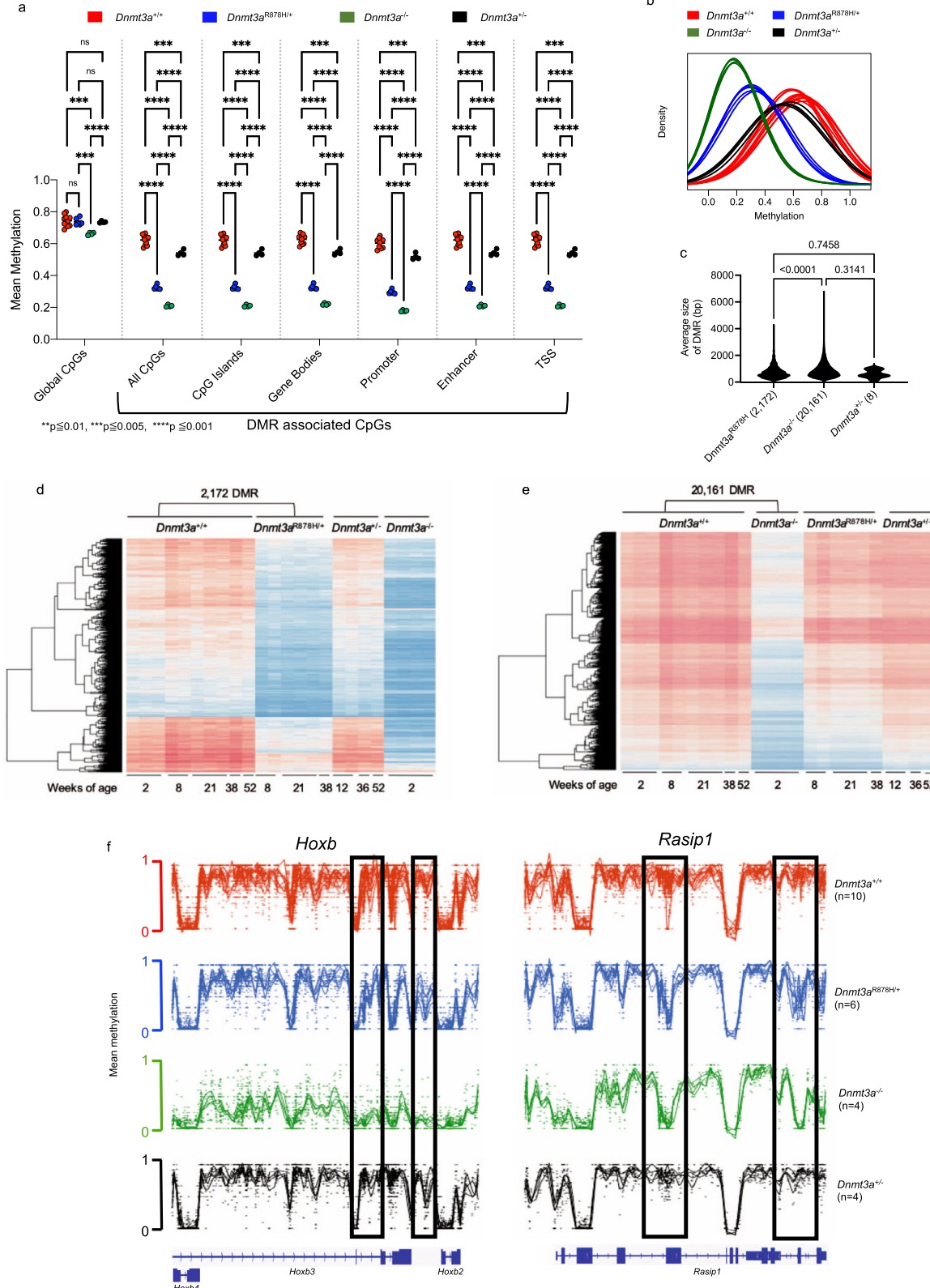

color flow cytometry, we verified this data in a larger cohort of mice, assessing both peripheral blood and bone marrow cells from $Dnmt3a^{+/+}$ and $Dnmt3a^{R878H/+}$ mice ranging from 1 month to 2 years of age (Supplementary Fig. 7). There were very few perturbations in mature cell populations (B-, T-, erythroid and myeloid cells), stem populations, or progenitor populations, although age-related alterations were observed for both genotypes.

We next asked whether $Dnmt3a^{R878H/+}$ derived hematopoietic cells would exhibit defects following the stress of a cytotoxic challenge with doxorubicin and cytarabine; however, there were no significant differences in count recovery for the $Dnmt3a^{R878H/+}$ mice (Fig. 7a). Regardless, spontaneous, fatal hematopoietic malignancies arose in six out of 80 unmanipulated $Dnmt3a^{R878H/+}$ mice after 1 year of age, vs. 0/65 WT mice ($p = 0.0296$; Fig. 7b). Flow cytometry

**Fig. 5 Germline _Dnmt3a_R878H/+ mice exhibit focal, canonical DNA hypomethylation in bone marrow cells. a** The mean methylation values for both global CpGs, and DMR-contained CpGs, in annotated regions of the genome are shown. Hypothesis testing was performed using two-way repeated-measures ANOVA with Tukey's multiple comparison test within each genomic region. **b** The density plot of methylation values from whole-genome bisulfite sequencing (WGBS) for differentially methylated regions (DMRs) defined by comparing _Dnmt3a_+/+ and germline _Dnmt3a_R878H/+ for each whole bone marrow sample from _Dnmt3a_+/+ (red, _n_ = 10), _Dnmt3a_R878H/+ (blue, _n_ = 6), _Dnmt3a_+/− (black, _n_ = 4) and _Dnmt3a_−/− mice (green, _n_ = 4). **c** The mean size (bp) for all DMRs identified in _Dnmt3a_R878H/+ (_n_ = 2,172), _Dnmt3a_+/− (_n_ = 8), and _Dnmt3a_−/− (_n_ = 20,161) bone marrow cells when independently compared to _Dnmt3a_+/+ controls (_P_-values by one-way ANOVA with Tukey's multiple comparisons test shown). **d** The heatmap showing mean methylation values for the 2172 DMRs defined in Panel _b_ for each individual _Dnmt3a_+/+ and _Dnmt3a_R878H/+ sample. Values for the same DMRs were plotted passively for _Dnmt3a_+/− and _Dnmt3a_−/− samples. **e** The heatmap showing mean CpG methylation values for the 20,161 DMRs defined by comparing the _Dnmt3a_+/+ and _Dnmt3a_−/−samples. Values for the same DMRs were plotted passively for _Dnmt3a_R878H/+ and _Dnmt3a_+/−. **f** Examples of _Dnmt3a_-dependent hypomethylated regions in the _Hoxb_ cluster (left) and the _Rasip1_ gene (right). The locations of DMRs in each gene are indicated in boxes. DMR differentially methylated region, bp base pairs, TSS transcriptional start site.

and morphologic examination using the Bethesda criteria[35,36] by a board-certified hematopathologist were used to classify two samples as MDS with maturation (mLeuk1 and mLeuk2), two as B-cell malignancies with extensive plasma cells in the bone marrow and spleen (mLeuk3 and mLeuk4), one as AML without differentiation (mLeuk5), and one as CMML-like (mLeuk6) (Fig. 7c and d).

## Discussion

In this report, we describe DNA methylation alterations and their consequences in human patients, and a mouse model of the _DNMT3A_ Overgrowth Syndrome (DOS). The peripheral blood of DOS patients had focal, canonical DNA hypomethylation that was more severe in patients with mutations that altered amino acid R882, but present in all patients with non-R882 mutations. In mice with a germline _Dnmt3a_R878H/+ mutation, we found similarities to human patients for methylation and gene-expression patterns, as well as similar growth and behavioral alterations, strongly suggesting that this mutation can cause the syndrome. This model supports the observation that patients with clonal hematopoiesis caused by mutations in _DNMT3A_ can live for many years without clinical progression to AML[14–16,37]. Similar to patients with clonal hematopoiesis, some mice with _Dnmt3a_R878H/+ develop spontaneous hematologic malignancies after long latent periods.

All of the germline _DNMT3A_ mutations examined in this study caused a focal methylation phenotype in the hematopoietic cells of DOS patients, suggesting that DOS-associated _DNMT3A_ mutations must cause a loss of DNA methyltransferase activity. However, the hypomethylation phenotype of patients with R882 mutations was much more severe than that of non-R882 mutations, consistent with the observation that R882 mutations cause a dominant-negative effect in hematopoietic cells[20,22]. The patients with _DNMT3A_R882 mutations had a total of 2,209 DMRs identified in their peripheral blood cells, while patients with non-R882 mutations had one-tenth that number. By coincidence, one patient had a true haploinsufficient state (UPN 518693), due to a heterozygous ~135 kb deletion that encompassed the entire _DNMT3A_ gene (Table 1, Supplementary Fig. 1). This patient's methylation phenotype defined the consequences of simple haploinsufficiency due to gene deletion. Since all of the other non-R882 missense mutations had a similar magnitude of methylation loss (Fig. 1), we suggest that the non-R882 mutations in this study may all lead to functional deficits that mimic the inactivation of the affected allele. However, the mechanisms of inactivation are almost certainly multifarious. The canonically hypomethylated DMRs in the non-R882 patients affected many of the same regions as in the R882 samples; 215/332 DMRs (65%) were concordant, suggesting that these regions may be highly relevant for epigenetic changes that alter the state of hemato-poietic stem cells, rendering them more susceptible to transformation.

As noted in similar studies[22,29], the global correlation between DMRs and altered gene expression is relatively weak in the hematopoietic cells of the DOS patients. However, two example genes highlighted in this study (_HOXB2_ and _RASIP1_) had reproducible but opposite alterations in gene-expression patterns that were associated with local DMRs. _Hoxb4_ expression and methylation correlations have previously been reported in a _Dnmt3a_ deficient serial transplantation model[32]. Globally, how-ever, no strong correlative rules could be established for the expression of genes in close proximity to DMRs in peripheral blood cells, even when restricted to specific functional regions in the genome (Supplementary Fig. 2). Similar findings were reported in a parallel yet unique model of complete _Dnmt3a_ loss[29]. Clearly, the expression of genes near DMRs are strongly influenced by other factors, including genomic context (exon vs. intron, early vs. late exon, etc.), transcription factor and protein networks, chromatin modifications, and long-range DNA inter-actions, none of which have yet been adequately explored in an integrated analysis.

The germline _Dnmt3a_R878H/+ mouse model phenocopies many key aspects of the human syndrome. Age-related weight gain was associated with increased body fat content (with no change in lean mass), suggesting that these mice were indeed obese; obesity was exacerbated with high-fat diet but was not associated with overeating, and its physiologic basis was not revealed by standard metabolic studies. Epigenetic variability is associated with human obesity and can be altered by inheritance, diet, aging, and the intrauterine environment[38]. Behavioral tests revealed that _Dnmt3a_R878H/+ mice had reduced exploratory behavior, volitional movement deficits, and complex changes in learning and memory or anxiety-like behaviors often observed in DOS patients. While movement, coordination, memory, learning, and anxiety are intertwined, untangling these relationships in future studies will be important for understanding specific con-tributions of individual mutations to phenotypes in DOS patients. This model shows more locomotive deficits and less autism-like behaviors than noted in haploinsufficient mice[39], suggesting possible specific genotype: phenotype relationships for different _Dnmt3a_ mutations. Furthermore, movement deficits, which may be a consequence of behaviors, may contribute to the weight phenotype (or vice versa), and therefore, may have important implications for the development of obesity in DOS patients. Further studies of brain methylation phenotypes will be required to fully elucidate epigenome: phenotype correlations in these two models of DOS.

There are a number of epigenetic similarities between DOS patients and mice with germline _Dnmt3a_R878H/+ mutations. We identified 2172 DMRs in the bone marrow cells of _Dnmt3a_R878H/+ mice, while 2209 were identified in the blood cells of _DNMT3A_R882 DOS patients. Because the age and sex of the samples from both species were well matched, it is very unlikely

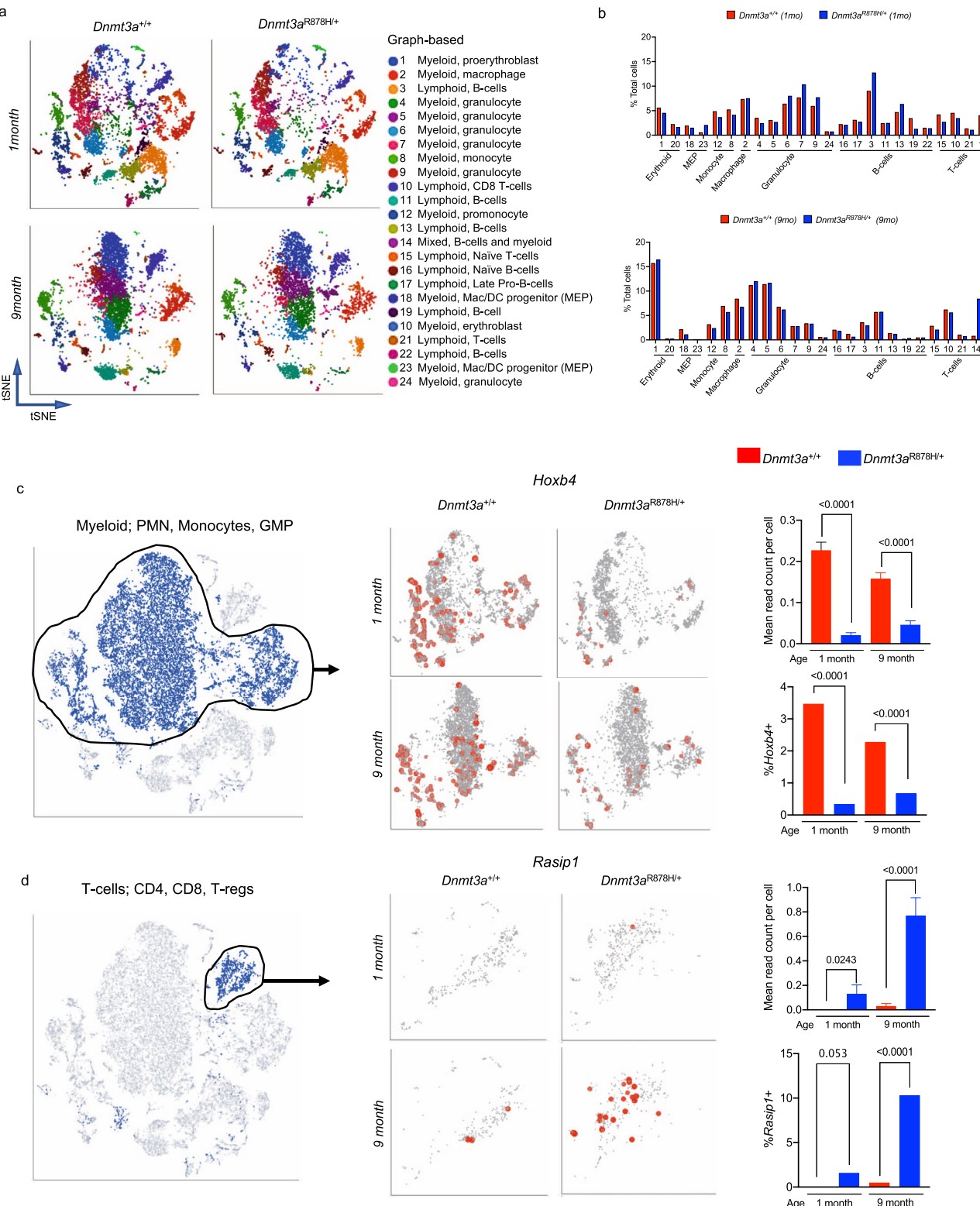

**Fig. 6 Germline *Dnmt3a*^R878H/+ mice have differentially expressed genes in hematopoietic cells. a** tSNE projections of scRNA-seq data from whole bone marrow samples from *Dnmt3a*^+/+ (left, *n* = 2) and *Dnmt3a*^R878H/+ (right, *n* = 2) mice at 1 month and 9 months of age, showing known populations defined by graph-based clustering and defined by ToppGene. **b** Population fractions associated with scRNA-seq data shown in Panel *a* by genotype and age. **c** and **d** Examples of differentially expressed genes identified in both humans and mice, and in more than one graph-based cluster, and associated with a differentially methylated region; tSNE projection, read counts per cell in all cells, and percentage of myeloid cells expressing *Hoxb4 (panel c)* and T-cells expressing *Rasip1 (Panel d)* are shown (Data are presented as mean values +/- SEM. *P*-values by two-tailed *t* test or Fisher's exact test for ratios are indicated, *n* = 2 biologically independent samples per genotype). PMN polymorphonuclear leukocyte, GMP granulocyte-monocyte precursor.

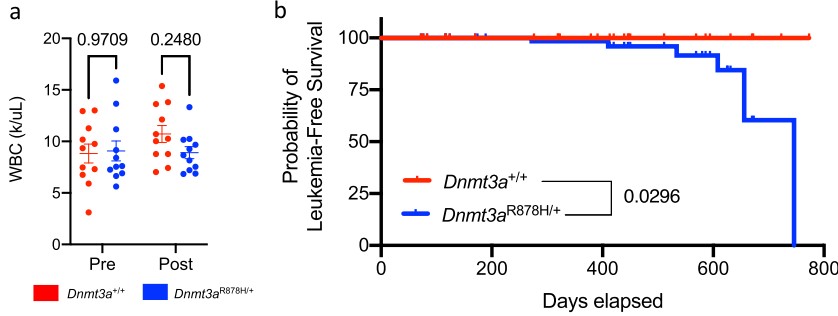

| Mouse ID | Age (y) | Spleen wt (grams) | Bethesda Criteria | Hb (g/dL) | PLT (K/uL) | WBC (K/uL) |
|---|---|---|---|---|---|---|
| mLeuk1 | 1.67 | 1.14 | MDS with maturation | 12.5 | 1022 | 6.04 |
| mLeuk2 | 1.46 | 0.66 | MDS with maturation | 7.3 | 1804 | 5.16 |
| mLeuk3 | 1.12 | 0.3 | B-cell | 9.9 | 1287 | 9.9 |
| mLeuk4 | 1.37 | 0.3 | B-cell | 9.9 | 1572 | 29.14 |
| mLeuk5 | 2.04 | 1.06 | AML | NA | 957 | 6.86 |
| mLeuk6 | 1.80 | 2.06 | CMML-like | 11.1 | 975 | 11.26 |

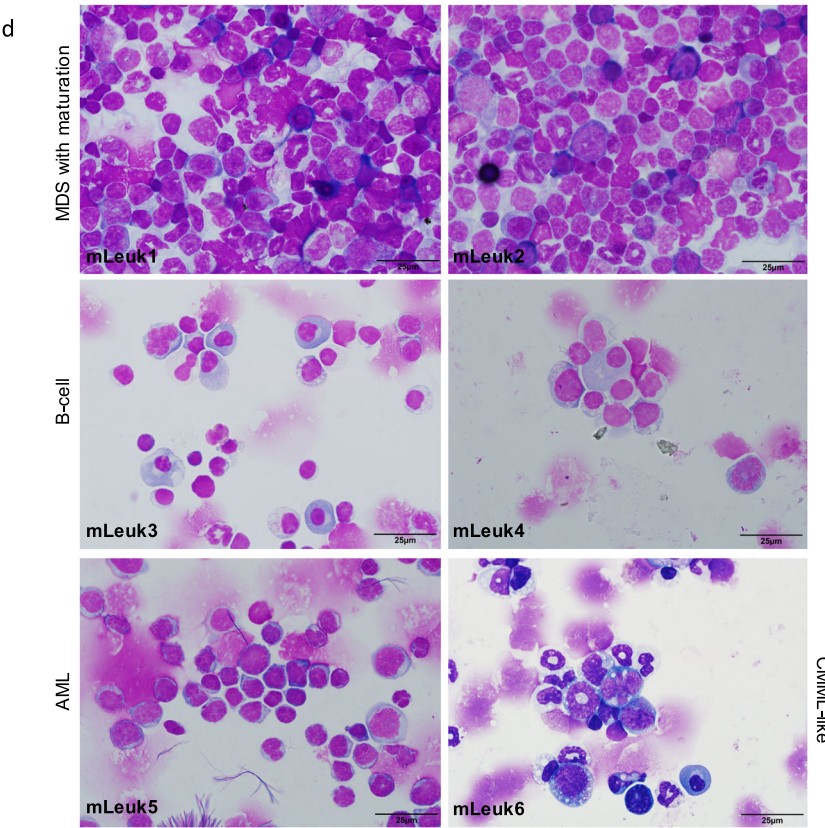

that these DMRs were significantly affected by either covariate. Further, a post hoc sensitivity analysis of defined DMRs using linear regression to test for the age and sex as covariates revealed that genotype remained the most important predictor of methylation phenotype. The frequency of DMRs in annotated genomic locations was similar in both species, for example, DMRs were found in 7.52% of the gene bodies of R882 mutant human blood cells, vs. 6.24% of gene bodies in mice with the R878H mutation (Supplementary Fig. 1). Although direct human to mouse synteny mapping outside of gene bodies is informatically challenging, we have found that many of the DMRs within specific genes are similar between humans and mice. For illustrative purposes, we have highlighted similarly located DMRs in the *HOXB* gene cluster and the *RASIP1* gene in both species. Although we do not

**Fig. 7 Germline _Dnmt3a_ R878H/+ mice develop hematopoietic malignancies. a** _Dnmt3a_+/+ (_n_ = 11) and _Dnmt3a_R878H/+ (_n_ = 11) mice were treated with 3 mg/kg doxorubicin for days 1–3 and 100 mg/kg cytarabine for days 1–5 administered by tail vein injection, and total peripheral blood white cell counts were assessed before and 24 days after chemotherapy. Individual data points are presented and mean values + /- SEM are shown. _P_-values were calculated by two-way ANOVA and Šídák's multiple comparisons test. **b** Kaplan–Meier curve of leukemia-free survival over time for _Dnmt3a_+/+ (_n_ = 65) and _Dnmt3a_R878H/+ mice (_n_ = 79). **c**) Characteristics of six spontaneous hematopoietic malignancies arising in _Dnmt3a_R878H/+ mice. **d** May–Grünwald Giemsa stained cytospins of spleen or bone marrow samples from spontaneous malignancies arising in _Dnmt3a_R878H/+ mice, as summarized in **c**. WBC (K/uL) white blood cells (x1000/mL), pre pre-chemotherapy, Post postchemotherapy, y years, Hb (g/dL) hemoglobin (grams/decilitre), PLT (K/uL) platelets (x1000/mL), MDS myelodysplastic syndrome, AML acute myeloid leukemia, CMML chronic myelomonocytic leukemia.

suggest that dysregulation of these specific genes is directly relevant for the DOS phenotype, it is interesting to note that _HOXB_ cluster genes have been found to be downregulated in children with autism spectrum disorders[40], and that the _RASIP1_ locus has been found to be hypomethylated in the blood of children with the Floating Harbor Syndrome[41], which is caused by pathogenic variants in _SRCAP_, and associated with facial dysmorphology and learning disabilities (MIM #136140). The similarity between phenotypes, DMRs, and DEGs between species suggests that this mouse model will be useful for the study of the epigenetic phenotypes in various organs, and how they may relate to the complex phenotypes observed in patients with this disorder.

Is DOS a leukemia susceptibility syndrome? Evidence from this and other studies suggest that it is[2,4,6,13]. Because many of the recently identified DOS patients are quite young, the lifetime risk of developing a hematologic malignancy is not yet clear. The incidence of childhood ALL is 35 per million, and childhood AML, 7 per million;[42] although somatic _DNMT3A_ mutations are common in adult AML, they are extremely rare in children with AML[43]. The number of identified DOS patients worldwide is currently ~200 (J. Kiernan, personal communication). Several of these patients have developed hematologic malignancies at an early age. In the largest series of DOS patients reported to date[2], one out of 55 had developed AML at age 12 (_DNMT3A_Y735S). In this series of 11 patients, one had presented with AML at age 12 (_DNMT3A_R882C); an additional DOS patient with a _DNMT3A_R882C mutation developed AML at age 15[4]. With the help of the TBRS Community (a group of concerned TBRS families that facilitates research for this syndrome; https://tbrsyndrome.org), we have identified several other children and young adults with DOS and a history of hematologic malignancies (Ferris, Smith, and Ley, manuscript in preparation), including a 20-year-old with AML (_DNMT3A_I310F), a 9-year-old with Pre-B-ALL (_DNMT3A_R882H), a 7-year-old with a secondary T-cell leukemia/lymphoma (_DNMT3A_R882H), a 34-year-old with Essential Thrombocytosis (_DNMT3A_R882H), and a 27-year-old with Hodgkin Lymphoma (_DNMT3A_Y735C). Importantly, cooperating mutations in genes identified in DOS patients with AML are similar to those found in adults with _DNMT3A_ mutant AML, including _FLT3_-ITD, _NPMc_, and _PTPN11_ (UPN 894912)[4]. Mouse models have further established the risk of developing hematopoietic malignancies: _Dnmt3a_ deficiency is associated with the development of myeloid, erythroid, B-, and T-cell malignancies[32,44–50] and _Dnmt3a_ haploinsufficiency in the germline is associated with the development of myeloid malignancies after a long latent period[30]. In this report, _Dnmt3a_R878H/+ germline mutations are associated with the development of spontaneous myeloid and B-cell neoplasms. Natural history studies in humans will be needed to define the relative risk of leukemia development for different mutation types (i.e., haploinsufficiency-like vs. R882 mutations), and the nature of the genetic and epigenetic events that provoke progression. Regardless, these data suggest that the children with this disorder will need to be prospectively monitored for the development of hematopoietic (and perhaps other) malignancies. Indeed, young DOS patients have been described with a pituitary macroadenoma[51],

a medulloblastoma[52], and a dermatofibrosarcoma in one patient in this study (UPN 228211).

DEGs defined in preleukemic blood cells in this study may provide some clues regarding the altered epigenetic state that predisposes to transformation. For example, several _HOXB_ cluster genes are downregulated in cells expressing _DNMT3A_R882H. However, in AML cells, the same genes are often persistently expressed at high levels[53], and genes within this cluster are normally repressed by methylation of the DERARE element between the _Hoxb4_ and _Hoxb5_ genes[54]. _HOXA_ and _B_ cluster genes can also be dysregulated in AML by fusions with _NUP98_[55–61], and dysregulation is seen in association with MLL translocations[62] and _NPM1_ mutations[63]. These observations underlie the complexity of _HOX_ gene dysregulation patterns in AML and underscore the large gaps in our knowledge regarding the events that govern the progression from preleukemia to overt transformation.

In summary, the data presented in this report suggest that many _DNMT3A_ mutations associated with DOS decrease the methyltransferase activity of DNMT3A, causing a focal, canonical hypomethylation phenotype. This phenotype is more pronounced with mutations at R882, probably because of the dominant-negative effects of these mutations. The germline R878H mutation results in a mouse phenotype that recapitulates many features of human patients with the same mutation, strongly suggesting that this mutation is sufficient to cause the syndrome. The availability of this model will allow for more precise characterization of the epigenetic states that are responsible for the various features of this syndrome, and should provide a preclinical model for approaches designed to correct it.

## Methods

**Primary human peripheral blood samples**. All primary _DNMT3A_ overgrowth syndrome samples were from buffy coat preps from the peripheral blood samples shipped overnight to Washington University School of Medicine, or obtained locally. Participation in the study was governed by informed consent obtained from the parents or guardians of the patients, using protocols that were compliant with specific ethical regulations for human studies, in accordance with the Declaration of Helsinki. The consent documents and protocol were approved by the Human Research Protection Office at Washington University School of Medicine (Protocol #201011766). Patient and/or family consent included the collection and sharing of anonymized data with the scientific community, including (but not limited to) genetic information, nonidentifying personal characteristics (including age, sex, and ethnicity), and other clinical characteristics. Confirmation of germline _DNMT3A_ mutations was performed in peripheral blood and buccal swab DNA with whole-exome sequencing. Of note, the fresh sample utilized for scRNA-seq from UPN 624400 was obtained after the patient had been treated for acne using isotretinoin for approximately 3 months.

**Generation of mouse model and husbandry**. All mice were in the C57BL/6 J background. Mice with the germline, heterozygous _Dnmt3a_R878H/+ mutation (mouse homologue of the human _DNMT3A_R882H) from the endogenous locus were produced by crossing the conditional model first described by Guryanova et al.[28] with B6.C-Tg(CMV-cre)1Cgn/J mice (The Jackson Laboratory). A construct containing a tetracycline-inducible, WT human _DNMT3A1_ cDNA was introduced into the C57Bl6/J genome, and founders were crossed with R26-M2rtTA mice in the B6 background as described in Ketkar et al.[29]. All mouse studies were performed in accordance with institutional and federal guidelines, were compliant with all ethical considerations for animal studies, and were approved by the Animal Studies Committee at Washington University (Protocol 20–0053. Mice were

housed at 20 degrees Celsius, 65% humidity and dark/light cycle from 5am–7pm. Peripheral blood was obtained from the retroorbital space with EDTA capillary tubes (Fisher Scientific) after adequate anesthesia with isoflurane. Femur, tibia, and pelvis-derived bone marrow cells were collected from mice in PBS supplemented with 2% FBS (Atlanta Biologicals) and 0.5 mM EDTA pH 8.0. Cell pellets were treated with Ammonium Chloride/KCl (ACK) red cell lysis buffer and resuspended in the RPMI/FBS mixture as a working solution. Female mice were utilized for high-fat diet feeding for weight tracking due to the propensity for male C57Bl/6 mice to gain weight[38].

**Flow cytometry mouse.** The peripheral blood and bone marrow from mouse femurs and tibias were utilized for flow cytometry applications. Once isolated, cells were treated with ACK red cell lysis buffer. Cells were stained with combinations of the following antibodies against cell-surface markers to identify indicated cell types (all antibodies are from Becton Dickinson unless indicated): CD11b (MI/70, 0.125 µL), Gr-1 (TB6-8C5; Biolegend, 0.5 µL), Ter-119 (Ter119, 0.25 µL), CD71 (C2, 0.5 µL), B220 (RA3-6B2, 1 µL), CD19 (6D5, Biolegend, 0.25 µL), CD3e (145-2C11, 1 µL), NK1.1 (PK136, 0.5 µL) Sca-1 (D7; Biolegend, 0.0625 µL), c-KIT (2B8, 1 µL), CD34 (RAM34, 2 µL), FLT3 (A2F10; Biolegend, 0.5 µL), CD150 (TC15-12F12.2; Biolegend, 0.5 µL), CD48 (HM48-1; Biolegend, 0.0625 µL), Ly5.1 (A20, 0.8 µL), Ly5.2 (104, 1 µL). The following phenotypes were used to define stem and progenitor populations: lineage negative (Lin): CD11b, Gr1, Ter119, CD71, B220, CD3e, NK1.1; LSK: Lin neg, Sca-1, c-KIT; GMP: Lin neg, Sca-1, c-KIT, CD34, FLT3; CMP: Lin neg, Sca-1, c-KIT, CD34, FLT3 neg; MEP: Lin neg, Sca-1, c-KIT, CD34 neg, FLT3 neg; SLAM: Lin neg, Sca-1, c-KIT, CD150, CD48 neg. Human. Peripheral blood from unaffected controls and DOS patients were utilized for flow cytometry applications. Antibodies against well-characterized cell-surface receptors were used to identify indicated cell types (all antibodies are from Becton Dickinson unless indicated; clone indicated in parentheses): CD45 (HI30, 2 uL), CD3 (UCHT1, 1 uL), CD11b (MI/70, 0.125 µL), CD16 (3G8; Biolegend, 0.5 µL), CD19 (HIB19, Biolegend, 1 µL), CD14 (M5E2, BioLegend, 1 µL), HLA-DR (G46-6, 1 µL), CD15 (HI98, 1 µL), CD56 (NCAM16.2, 0.5 µL), CD33 (WM53, 0.1 µL), CD11c (B-ly6, 0.125 µL), CD66b (G10F5, BioLegend, 0.5 µL), CD123 (32703, R&D systems, 0.25 µL), CD303 (REA693, Miltenyi, 0.5 µL). Acquisition was performed using the ZE5 flow analyzer (Bio-rad) and data were analyzed using FlowJo (Tree Star) and Prism 9 (GraphPad). Gating strategies are shown in Supplementary Fig. 9.

**Whole-genome bisulfite sequencing.** WGBS was performed using the AccelNGS Methyl-Seq DNA library kit (Swift Biosciences, #30096). The sequence was generated on Illumina HiSeq or NovaSeq instruments and reads were mapped with biscuit, as described in https://github.com/genome/analysis-workflows/blob/v1.5.0_fix_1/definitions/pipelines/bisulfite.cwl. The "metilene" algorithm was used to analyze the methylation ratios at all CpGs to identify differentially methylated regions (DMRs). Each DMR was required to span >10 CpGs, have a mean methylation difference between groups of >0.2, and a false discovery rate (FDR) <0.05. Adjacent DMRs <50 bp apart were merged, then non-canonical DMRs with standard deviation within either group of >0.1 were removed[24]. Gene annotations (gene bodies, transcription start sites, promoters, and enhancers) are based on protein-coding genes from Ensembl release 95 and its accompanying regulatory build.

**Total RNA-sequencing, processing, and analysis.** Total RNA-sequencing of *DNMT3A*[R882] and healthy control peripheral blood was performed using the Illumina TruSeq Stranded Total RNA Library Kit[29]. Alignments were generated with HISAT as detailed in https://github.com/genome/analysis-workflows/blob/v1.5.0_fix_1/definitions/pipelines/rnaseq.cwl. Expression quantification was done using Partek Flow software (version 9.0.20.1125). Differentially expressed genes were identified using DESeq2 between *DNMT3A*[+/+] vs *DNMT3A*[R882], requiring a p-value of ≤ 0.05 and a Log2Ratio of ≥1 or ≤−1.

**Single-cell RNA library construction and sequencing.** Cells were processed using the 10x Genomics Chromium Controller and the Chromium Single Cell 5′ Library & Gel Bead Kit (v2.0) following the standard manufacturer's protocols (https://tinyurl.com/y96l7lns). Samples were then sequenced on the Illumina NovaSeq (2 × 150 paired-end reads), yielding a mean of 486,655,675 reads per sample (Dataset S8). Transcript alignment and counting were performed using the Cell Ranger pipeline (10x Genomics, default settings, Version 3.0.1). Using a nearest neighbor algorithm implemented in R (version 3.5.1), cells were annotated according to the Haemopedia expression atlas of hematopoietic cell types[34] using code available here: (https://github.com/genome/docker-scrna_lineage_inference). Using Partek Flow software (version 9.0.20.1125), we removed cells that contained fewer than 250 expressed genes, less than 500 total reads, or more than 10% mitochondrial transcripts. For each cell, the expression of each gene was normalized to the sequencing depth of the cell, scaled to a constant depth (10,000), and log-transformed. Dimensionality reduction and visualization were performed with the t-SNE algorithm. Differentially expressed genes were determined using the ANOVA algorithm and defined as having an *p*-value of ≤0.05 and a Log2Ratio of ≥ 1 or ≤−1.

**Body composition.** The fat mass and lean mass of 10 pairs of live *Dnmt3a*[+/+] vs. *Dnmt3a*[R878H/+] mice were measured using whole-body quantitative magnetic resonance with an EchoMRI Body Composition Analyzer as described in Nixon et al.[64]. Briefly, animals were placed in a thin-walled plastic cylinder with a solid cylindrical plastic insert added to limit movement. Animals were then subjected to a low-intensity electromagnetic field that generates a signal that modifies the spin patterns of hydrogen atoms within tissues, the relaxation curves of which are altered depending on the content of fat and lean tissue. Canola oil was used as the standard for measurements.

**CT scans.** CT scans of four pairs of *Dnmt3a*[+/+] vs. *Dnmt3a*[R878H/+] mice at 30 weeks of age were taken on a Siemens Inveon MM PET/CT scanner and images were acquired with the Siemens Inveon Research Workstation. All studies were conducted in the MIR Preclinical Imaging Facility of the Washington University School of Medicine.

**Craniofacial data collection and analysis.** We collected 25 landmarks from the reconstructions of cranium and mandible of WT and R878H mice using Stratovan checkpoint (www.stratovan.com). Two data collection trials were collected for each mouse and the average of those trials was used in the analyses.

These landmarks were used in two separate analyses: generalized Procrustes analysis (GPA)[65,66] and Euclidean Distance Matrix Analysis (EDMA)[67,68]. The cranial landmarks were analyzed separately from the mandibular landmarks. GPA was used to complete principal components analysis (PCA), which determines whether the two groups are different without allowing us to localize the changes. With GPA, the landmarks are scaled, rotated, and translated to achieve the best fit. To statistically test for localized differences between the groups of mice for linear distances, Euclidean Distance Matrix Analyses was used. EDMA statistically compares linear distances between pairs of landmarks to determine whether the mice are different for these distances.

**Femoral data collection and analysis.** Three-dimensional reconstructions of the mouse femora were completed using Avizo (https://www.thermofisher.com). Using the Measure tool in this software, the length of the left femur for each mouse was collected twice. Femur length was calculated as the distance spanning between the greater trochanter and lateral condyle. The average of the two trials was used to compare overall femur lengths between the *Dnmt3a*[+/+] vs. *Dnmt3a*[R878H/+] mice with a Mann–Whitney U-test.

**Behavior measurements.** All behavioral analyses were performed in collaboration with the Intellectual and Developmental Disabilities Research Center at Washington University School of Medicine in St Louis and were recently described in full by Christian et al[39]. Mouse numbers used for each significant finding are indicated in Fig legends. One-hour locomotor activity was evaluated by computerized photobeam instrumentation in transparent polystyrene enclosures (47.6 cm × 25.4 cm × 20.6 cm)[69]. Activity variables such as ambulation's and vertical rearing's were measured in addition to time spent in a 33 cm × 11 cm central zone[69]. Sensorimotor battery assayed walking initiation, balance (ledge and platform tests), volitional movement (pole and inclined screen test), and strength (inverted screen) were previously described[69,70]. For the walking initiation test, mice were placed on the surface in the center of a 21 cm × 21 cm square marked with tape and the time for the mouse to leave the square was recorded. During the balance tests, the time the mouse remained on an elevated plexiglass ledge (0.75 cm wide) or small circular wooden platform (3.0 cm in diameter) was recorded. For the inclined screen tests, a mouse was placed (oriented head-down) in the middle of an elevated mesh grid measuring 16 squares per 10 cm angled at 60° or 90°. The time for the mouse to turn 180° and climb to the top was recorded. For the inverted screen test, a mouse was placed on a similar screen and when it appeared to have a secure grasp of the screen, it was inverted 180° and the latency to fall was recorded. All tests had a duration of 60 s, except for the pole test which was 120 s. Two separate trials were done on subsequent days, and the average time of both trials was used for analysis. *Morris water maze.* Cued trials (visible platform, variable location) and place trials (submerged, hidden platform, consistent location) were conducted in which escape path latency, length, and swimming speeds were recorded. Animal tracking was done using a computerized system (ANY-maze, Stoelting). During cued trials, animals underwent 4 trials per day over 2 consecutive days with the platform being moved to a different location for each trial, and few distal spatial cues. Each trial lasted no longer than 60 s, with a 30-minute interval between each trial. Performance was analyzed across four blocks of trials (two trials/block). After a three-day rest period, animals were tested on Place trials, in which mice were required to learn the single location of a submerged platform with several salient distal spatial cues. Place trials occurred over 5 consecutive days of training, with two blocks of two consecutive trials (60 s trial maximum, 30 s interatrial-interval after the mouse has reached the platform), and with each block separated by 2 h. Mice were released into different quadrants over different trials. Place trials were averaged over each of the five consecutive days (four trials/block). One hour after the final block, a Probe trial (60 s trial maximum) in which the platform is removed, and the mouse is released from the quadrant opposite where the platform had been located. The time spent in pool quadrants and the number

of crossings over the exact platform location were recorded. *Elevated plus maze.* Anxiety-like behaviors were examined using the elevated plus maze[71]. The apparatus contains a central platform (5.5 cm × 5.5 cm) with two opposing open arms and two opposing closed arms (each 36 cm × 6.1 cm × 15 cm) constructed of black Plexiglas. Mouse position is measured using beam-breaks from pairs of photocells configured in a 16 × 16 matrix, and outputs are recorded using an interface assembly (Kinder Scientific) and analyzed using software (MotoMonitor, Kinder Scientific) to determine time spent, distance traveled, and entries made into open arms, closed arms, and the center area. Test sessions were conducted in a dimly lit room with each session lasting 5 min, and each mouse tested over three consecutive days. *Conditioned Fear*[72,73]. Mice were habituated to an acrylic chamber (26 cm × 18 cm × 18 cm) containing a metal grid floor and an odorant, all light with LED light which remained on for the duration of the trial. Day 1 testing lasted 5 min in which an 80-dB tone sounded for 20 s at trial timepoints 100, 160, and 220 s. A 1.0 mA shock (unconditioned stimulus) occurred within the last 2 s of the tone (conditioned stimulus). Baseline freezing behavior during the first 2 min and the freezing behavior during the last 3 min was quantified using image analysis (Actimetrics, Evanston, Illinois). On Day 2, testing lasted for 8 min in which the light was illuminated but no tones or shocks were presented. On day 3, testing lasted for 10 min in which the mouse was placed in an opaque chamber with a different odorant than the original test chamber. The 80-dB tone began at 120 s and lasted for the remainder of the trial, and freezing behavior to the conditioned auditory stimulus was quantified for the remaining 8 min. *Continuous and Accelerating Rotarod.* Motor coordination and balance was assessed using the rotarod test (Rotamex-5, Columbus Instruments, Columbus, OH) using three conditions: a stationary rod (60 s maximum), a rotating rod at constant 5 rpm (60 s maximum), and a rod with accelerating rotational speed (5–20 rpm, 180 s maximum)[70]. This protocol is designed to minimize learning and instead measure motor coordination, so testing sessions were separated by 4 days to allow for extinction. Testing included one trial on a stationary rod and two trials on both the constant-speed rotarod and accelerating rotarod. *Marble Burying.* Marble burying is a natural murine behavior and has been used to indicate repetitive digging behavior as well as anxiety-related behaviors. The protocol was adapted from[72,74]. Marble burying assays occurred when the mice were 8 weeks of age. Mice were placed in a transparent enclosure (28.5 cm × 17.5 cm × 12 cm) with clean aspen bedding and 20 dark blue marbles evenly spaced in a 4 × 5 grid on top of the bedding. Animals were allowed to explore freely for 30 min, and the number of buried marbles were counted every 5 min by two independent observers. Marbles were considered buried if they were at least two-thirds covered by bedding. Between animals, enclosure, and marbles were cleaned thoroughly.

**Plasma glucose, lipid, and leptin measurements**. Mice were fasted for 6 h before isolation of plasma. Glucose was measured utilizing Autokit c, glucose diagnostic reagent (439–90901) from Wako Chemicals USA Inc., according to manufacturer's instructions. Total Cholesterol was measured utilizing Infinity Total Cholesterol kit (NC9343696) from Fisher Scientific. Free fatty acids were measured utilizing reagents (999–34691, 995–34791, 991–34891, 993–35191, 276–76491) from Wako chemicals USA Inc. Leptin was measured using Mouse/Rat Leptin Quantikine ELISA kit (MOB00B) from R&D systems according to manufacturer's protocol.

**In vivo chemotherapy treatments**. Before treatment, mice were retro-orbitally bled and a complete blood count (CBC) was performed. Mice were treated with 3 mg/kg doxorubicin for days 1–3 administered through tail vein injection and 100 mg/kg cytarabine for days 1–5 administered through tail vein injection. Mice were retro-orbitally bled to obtain post-treatment CBCs 24 days after the last dose was administered.

**Quantification and statistical analysis**. Statistical analysis was performed using either the software indicated above or in R (4.0.2). Hypothesis testing in *R* was performed using chi-square tests for categorical variables, *t* tests for continuous variables, Wilcoxon signed-rank tests for comparing means of gene-expression quartiles, or Fisher's exact test for differences between proportional data (i.e., lineage fractions). All reported significance metrics are corrected for multiple testing by Benjamini–Holm (FDR), Bonferroni or Yates methods (*p*-values), unless otherwise noted.

**Reporting Summary**. Further information on research design is available in the Nature Research Reporting Summary linked to this article.

## Data availability

Sequencing data for all mouse datasets were deposited to the NCBI, [https://www.ncbi.nlm.nih.gov/bioproject/PRJNA722276], and are available without restrictions. Sequencing data for all human datasets were deposited to dbGaP study # phs000159. The human datasets are available only via this controlled access database because patient identity can be ascertained from SNPs in whole-genome bisulfite and exome sequencing data; the IRB that approved this study required this caveat. Applications for access can be submitted via the dbGaP page. Source data are provided with this paper.

## Code availability

All pipeline code used for genomic data processing is available at [https://github.com/genome/analysis-workflows] (commit id: 174f3b2). Alignment and variant calling from genomic DNA are under pipelines/somatic_exome.cwl, alignment and expression quantification is under pipelines/rnaseq.cwl, and bisulfite alignment and methylation inference is under pipelines/bisulfite.cwl.

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

## Acknowledgements

The authors thank the TBRS Community, and especially Jill Kiernan and Kerry Grens, for help in identifying the subjects for this study, and the patients and families themselves for their participation. We also thank all referring physicians for providing records and obtaining blood samples for this study. This work was supported by an ASH Fellow to Faculty Award (AMS), NIH grants CA101937 and CA197561 (TJL), CA211782 (CAM), CA211466 (MPR), MH117405 (HWG), and the Barnes Jewish Hospital Foundation (TJL). We thank Dr. Eric Duncavage for providing Bethesda classifications for the mouse malignancies, Dr. David Wozniak for performing the animal behavioral studies, Ms. Mieke Hoock for excellent animal husbandry, and Drs. Olga Guryanova and Ross Levine for providing mice with the Lox-stop-Lox *Dnmt3a*^R878H allele.

## Author contributions

A.M.S. and T.J.L. designed research; A.M.S., T.A.L., M.P.R., N.M.H., S.A., D.L.C., and J.R.M., performed research; C.A.H., N.M.K., H.W.G. and C.F.S. contributed new reagents/analytic tools; A.M.S., S.M.R., H.J.A., C.A.H., N.M.K., M.P.R., D.L.V., C.A.M., and T.J.L. analyzed data; M.S., S.E.H., F.F., and D.Y.C. provided clinical advice/services; and A.M.S., and T.J.L. wrote the paper.

## Competing interests

The authors declare no competing interests.
