## [Peer Review File · Nature Communications]

REVIEWER COMMENTS

Reviewer #1 (Remarks to the Author):

Smith and colleagues explore the consequences of DNMT3A mutations in patients with DOS syndrome and present a mouse model for one mutation found in DOS. The overall conclusions are that the R882 mutation presents a more severe mutation type than others which is manifest in a greater loss of DNA methylation. A great deal of ancillary data supports these overall conclusions.

While the data are of interest to specialists in the field, the figures are over-packed and poorly described, making extraction of the messages very difficult. At a minimum, the figures and data presentation need an overhaul so that the work is more accessible to the audience of Nature Communications readers. Also, the authors also must perform some actual validations of the data they present to support the claims. Many concerns are cited below. Many are concerned with data presentation that will make it easier for the reader to view and understand the data.

Concerns (no particular order of importance):

1. The authors claim on page 3 that R882 is “the most common” site of missense mutation in DOS. While technically true, given that only about 100 patients world-wide are published, and as yet, only a very small number have R882, it seems a bit of an overstatement to claim that it is “common”.
2. There are many cases of mis-citing of papers or instances where other papers should be cited. Page 3 and 14, the authors cite Spencer et al in support of R882 being dominant negative, but Kim et al (Blood, 2013) were the first to show this and should also be cited. I don't think citation (32) supports the dominant-negative hypothesis. In addition, Several other papers previously showed the loss of DNA methylation promoters, enhancers, gene bodies etc in mouse and human samples- both reduced methylation and the enlargement of regions. These include Tovy (Cell Stem Cell, 2020- human cell line), Jeong (Nature Genetics, 2014, mouse, Jeong, Cell Reports). The authors suggest on page 10 they are the first to look at WGBS in the context of “deficiency”, but such data have been around for years in other papers, albeit not from a germline model. At least this prior work could be cited, particularly given that the overall observations concerning DNA methylation changes are essentially the same.
3. On page 6, the authors note a difference between patient and sibling of NK and T cell numbers and comment on “a potential lineage switch within this compartment”. The results section is no place for wild speculation based on one patient. Also, NK cells are not considered part of the same “compartment” with classic T-cells.
4. Page 9 refers to figure 5, but actually I believe this should be referring to figure 4.
5. The authors use single-cell sequencing extensively to look at mouse and human samples. However, the authors should be more circumspect when drawing conclusions about gene expression differences. For example, the authors make claims about gene expression of HoxB genes and Rasip1. The main figures clearly show that a larger number of cells have detectable transcripts in some samples, which the authors claim indicates the genes are expressed at higher or lower levels (the data shown don't support this). The supplement shows a graph with “LSmeans” on the Y-axis, but this term is not explained. It is presumably least-squares mean which can be used in some contexts to infer gene expression. To claim gene expression differences like this repeatedly and to justify amplifying the observation in the discussion, the authors should perform real-time PCR.

Figures:

Most main figures suffer from severe overcrowding and placement in the PDF that resulted in their being cut off on the right and sometimes the top. Many of the panels have text that is far too small, or features that are far too small. While readers can blow up images online, many readers

still print and read on paper, so the presentation should be legible without a magnifying glass. Some examples noted below.

Also, in general the supplemental information has negative data. I suggest eliminating panels that truly show no differences, and moving some of the information from the main figures into the supplement.

1. Figure 1- change color scheme in (a) to differ from that in (b) (confusing).
2. 1c- presentation of circles makes it difficult to distinguish the groups and the significance below could be lined up.
3. 1d- impossible to read the Y-axis
4. 1f and g- somewhat redundant – move one to supplement.
5. 1h/I are not well labelled. What are the green bars? Not indicated in legend. Difficult to discern the claimed DNA methylation difference in the *hoxB* cluster.
6. Figure 2- most panels the fonts are far too small. (c) very difficult to read gene names. (f) contains a black bar to the left which appears to be gene clusters. The red squares are difficult to glean information from. Only informative panel (that is readable) is (h). Is expression detected at all in the controls?
7. Figure 4- (k) is not listed in the legend and I guess it is redundant with (b).
8. Figure 6- (b) 2 colors of blue look the same. (c) many panels are uninformative- could show the ones that are relevant and move to supplement or make a table. Not much to see in these panels except the number of genes. (d) is not well explained at all. There are green boxes and blue boxes. They refer to “violin” plots which are nowhere to be found. Be extreme scrutiny, it seems that the main feature of the graph is a few outliers at one extreme of violin plots that have been smashed against the origin. The graph does not show expression differences, but cells in which a signal for that gene is detected, not to be confused with expression level.
9. Figure 5- cant tell the difference between pink and red on the plots. And for all the panels, the presentation could be better. The entire figure does not show much new.
10. Supp fig 1- suffers from the same presentation problems as Fig 1. The text states there are no differences in global methylation at the indicated features, but there are p-values given which appear to be significant. Which is it? Ditto figure Supp 6 except the null does appear to have some lower methylation even at a global level.
11. Figure 8. (c) shows leukemia free survival. The WT mice show no apparent deaths out to 800 days. In most facilities, at least ½ of WT animals would be lost by this time point. Did the authors really have no deaths? Or were they just not leukemia deaths? Did they actually examine the cause of death of the WT and ascertain they were leukemia free? I buy the overall conclusions, but probably this could be presented differently to account for deaths of animals that were not able to be determined.
12. Figure 7- way way too much data to look at on the right with the small plots, and the text is illegible. What is this trying to show? What is new information here?
13. Supp Fig 7- confusing and LSmeans is not strictly quantification of gene expression. What are B and D trying to show?
14. All figures should be carefully re-examined for legibility and messaging.

Reviewer #2 (Remarks to the Author):

In the manuscript, Smith and colleagues correlate functional mutations in DNMT3A with DNMT3A Overgrowth Syndrome (DOS) and hematopoietic malignancies. The authors whole genome bisulfite sequencing on peripheral blood of DOS patients harboring the DNMT3A R882 (n=3) and DNMT3A non-R882 (n=8). This analysis identified subtle decreases in methylation at CPG islands and promoters, but not gene bodies or enhancers for both the R882 and non R882 mutants. In addition, the R882 mutants showed significantly more differentially methylated regions (DMRs) than non-R882 when compared to the DNMT3A^{+/+} samples, suggesting that the methylation phenotype is more severe in the mutant patients.

Next, the authors generated a germline heterozygous *Dnmt3a* R878H/+ mouse model to investigate the molecular mechanisms promoting the pathogenesis. The *Dnmt3a* R878H/+ mutant

recapitulated phenotypic symptoms observed in humans such as obesity, longer femur and humerus lengths, and macrocephaly associated to behavioral issues such as volitional movement deficits, and emotional and several cognitive impairment. WGBS in bone marrow cells collected from wild-type, Dnmt3a R878H/+ and Dnmt3a-/- mice demonstrated that an overlap in the DMR of Dnmt3a R878H/+ and Dnmt3a-/- samples. However, the average methylation difference was greatest in the ko cells. In addition, scRNAseq in whole bone marrow cells collected from Dnmt3a R878H/+ and WT counterparts young (1 month) and aged (9 months) mice showed enrichment for hematopoietic stem and progenitor cells (HSPCs) and monocytes in the old mutant mice. This analysis also showed downregulation of Hoxb4 in PMNs, GMPS, and monocytes from the Dnmt3a R878H/+ mutant mice compared to the wild-type. Finally, cytotoxic challenges using standard chemotherapy such as doxorubicin and cytarabine decrease in overall survival of the mutant mice. This is an interesting study analyzing the methylation profile of hematopoietic cells from DOS patients. The authors presented a novel mouse model to investigate the mechanisms of DOS. The manuscript includes compelling evidence of the fidelity of the mouse model to replicate observations made in human patients. Overall, experiments are well designed and analyzed. However, the single cell RNA-seq data (Figure 6) is poorly presented. Moreover, the statistical analysis applied in the single-cell RNA-seq to compare the cell distribution in groups which contain only one sample (or is it two?) is unclear. The authors should refer to previous studies on mouse bone marrow samples (Giladi A., et al 2018, Izzo F et., 2020...) for a better analysis and representation of the scRNA-seq data. The molecular phenotypes described in the hematopoietic cells are interesting.

Major comments:

- Figure 2: The statistical analysis of the single cell RNA-seq from the peripheral blood of the patient samples is questionable. The authors analyze two samples of the same patient (one fresh and one frozen) and compare to one healthy donor. Therefore, it is only one patient per group. Although an analysis of distribution of the populations per sample is interesting, applying an statistical analysis (as shown in 2d) in only one sample per group is not correct. Authors should show the distribution of the cells per sample in bar plots or in a pie representation rather than statistical analysis between groups containing only one sample.
- Figure 2c: A better strategy for cell calling including several genes per group should be included in the main or supplementary figure. For instance a heatmap representation of genes in each population may be a better approach to highlight the gene profile of each cell type. How do the authors explain the lack of monocytes? Are some of the monocytes included in the neutrophil cluster?
- Figure 2e and f: Regardless of the GO mentioned on the NK cells, did the authors find upregulation or downregulation of specific pathways in other cell types? Is there any metabolic gene dysregulated? Or any other gene that could be extrapolated to the overgrowth syndrome?
- Figure 3a, b, j, k: It seems that the differences in weight differences are more pronounced as the WT and mutant age. What is the effect of the mutation on metabolism? Is there an age dependent metabolic impact of the Dnmt3a R878H/+ mutant? Authors could analyze metabolic profiles in the serum of the mice.
- Figure 4: Did the authors collect brain H&E in samples from the mice? Does the brain of mutant mice show major structural abnormalities?
- Figure 5f & g: Why are there no Dnmt3a+/+ 52 weeks of age samples while there are samples for the Dnmt3a +/- mutants? A heatmap or barplot for mean methylation values comparing age-matched or cohorts with a similar age group would be more clear and easier to interpret than how the data is presented currently. Since the mutation seems to have some age dependent effect, separating the cohorts might be more clear as well.
- Figure 6a: an UMAP combining all the samples together should be included in the figure, together with the independent ones.
- Figure 6a: Did the authors perform unbiased clustering? Were the samples normalized to the number of cells for each condition? Do the mutant samples cluster more closely together? Do the mutants show the emergence of any mutant specific cluster in the different populations? A density plot might show differences in transcriptional expression between the different conditions.
- What are the highly expressed genes for each cell population? Cell-specific heatmaps should be included in this figure or supplementary.
- Figure 6b: statistical analysis between groups that include only one sample is dubious at best. How many samples and individual mice were actually used in Figure 6? At least 3 animals in each

timepoint should be used.

- Figure 6: What were the phenotypes of the mice when they were used for the experiment? What was the disease severity of the mice?
- Figure 8b: Do the authors have an explanation for the variability in the transplantation experiment? Was there an observed difference in the mice that expanded in the host vs the one that did not engraft?
- Error bars and/or Statistical analysis should be included in the graphs of the supplementary figures: 6, 7 and 8.

Minor comments:

- Introduction: What is the percentage of patients with DOS that harbor the DNMT3A R882 mutations? How many of these patients develop hematologic malignancies and is it the leading cause of mortality within DOS patients?
- Mice with germline Dnmt3a R878H/+ exhibit behavioral abnormalities: Text does not match Fig 5 a-k, but refer to Figure 4 and Supplementary Fig 5 a-n.
- Fig 8b Do each of the lines represent one sample or a cohort of samples? Please clarify.

Reviewer #3 (Remarks to the Author):

This is a very interesting study describing the DNA methylation phenotypes and its transcriptional effects in humans and mice with DNMT3A overgrowth syndrome (DOS). The data presented suggest that many DNMT3A mutations associated with DOS decrease the methyltransferase activity of DNMT3A, causing a focal, canonical hypomethylation phenotype that is most pronounced in DNMT3A R882 mutations, most commonly found in cancer.

Major comments

DNA methylation analysis:

The authors used n=11 DOS (or TBRS) cases ranging from 20 months to 36 years compared to n=6 young adult controls. The first concern with this study design is the sample size. N=11 TBRS cases should be sufficient to identify DNA methylation changes across the genome with large effect sizes. However, cases were separated into subsequent groups based on the presence or absence of an R882 variant (n=3 with R882 and n=8 without) and each group was compared to just 6 controls. This number of controls is smaller than what is optimal to identify DNA methylation signatures. Currently, 2-3 times the number of cases is the target for the controls. The age range of the controls, as compared to the cases, needs to be matched given the strong association between age and DNAm (especially with such a wide age range within the cases), i.e. the control age range should reflect the 20 month-36 year range of the cases. As well, in silico work-arounds such as adjustments for blood cell composition (estimated via DNA methylation) are not yet available for WGBS data, as they are for microarray data and this should be presented as a limitation of the study. Additionally, it is not clear that sex was taken into account for DMR identification (i.e. used as a covariate), which is necessary.

The authors state that "To date, the methylation phenotypes of DOS patients with non R882 mutations have not yet been described,". However there have been at least 2 papers that report DNA methylation profiles for non R882 variants as well as other variants -see Jeffries et al 2019, PMID: 31160375 and Aref-Eshghi et al 2020, PMID: 32109418.

In addition, it is important to note that Jeffries et al PMID: 31160375 have reported age acceleration in DNMT3A variants and that age acceleration is much higher with R882 mutations. Have the authors looked into the effect of DNA methylation aging in their cohort? Maybe using DNA methylation aging as a confounding variable can further support their results.

Figure 1a: It is highly recommended to add the mutation information for all variants in this figure and replace the sample numbers.

There is no detailed information about the mutations i.e. cDNA and protein information. Also, there is no information about the clinical phenotype of the 11 DOS patients. The authors refer to clinical information in table 1 but table 1 is missing from the manuscript.

No information is available about the three R882 mutations -are they all R882H or a mix of R882H,

R882 C etc.

The statement “Of these DMRs, 134 (72%) overlapped with DMRs identified in the DNMT3AR882 21 samples (where overlap required at least 1bp of shared 22 sequence), “ suggests the fact that these 2 identified DNA methylation phenotypes correspond to one DNMT3A DNA methylation profile and the 28% that did not overlap between the 2 could be the result of power issues/sample size difference between the 2 groups. Have the authors considered generating a DNA methylation profile for all DNMT3A variants?

“The fraction of T cells was reduced (46.04% to 11.30.29%, $p = 3.017e-13$), and the NK cell fraction was increased (3.37% to 14.88%, $p \leq 12.2.2e-16$), suggesting a potential lineage switch within this compartment.” The authors state that this analysis was done comparing the patient to his unaffected sibling who is 4 year older than the patient. Unfortunately, the authors did not provide the age of the affected sibling. It is important to know the age to better understand the shift in cell types reported here and if it is due solely to age difference rather than DNMT3A status. (The patients included in this study are as young as 20 months old).

Given the observed difference in the phenotype with aging in mice with a germline DNMT3A heterozygous variant, it is recommended that the authors discuss what is known about the clinical features of DOS in adult patients compared to children. How do the clinical features of their older patients compare to the younger ones?

Minor Comments:

The authors state “To determine whether DOS patients have DNA methylation changes in the genomes of 5 of their hematopoietic cells, we obtained peripheral blood and buccal swab samples from children and adults with DOS”. It is not clear if buccal samples were used for DNA methylation analyses as well.

Figure 1h and 1i can be supplemented with violin plots or boxplots to better represent the DNA methylation differences at the selected DMRs between the 3 groups.

With such small numbers and an unbalanced study design, it is difficult to know if any other reported differences between the different variant types are meaningful. The effect of the R882 variant does indeed appear dramatic on the heat map (1F); however, it is difficult to discern the effect of age and stochastic changes in the small sample size from those of the disorder. This issue extends to comparing the DMR severity and size features of R882 variants vs. non-R882 variants. Table 1 seems to be missing so can't assess sex, age, and clinical features in the R882 variants vs. non-R882 variants.

In general, because of the study design, more care should be taken in drawing conclusions. In the discussion, the authors state: “Since all of the 6 other non-R882 missense mutations had a remarkably similar methylation profile 7 (Figure 1g), we suggest that all non-R882 mutations essentially inactivated the affected allele”. The similarity of DNAm profiles cannot yet be used to make these kinds of determinations, even when statistical methods are more rigorous than those used here. This represents an over-interpretation of the data.

The identification of changes in genes such as HOX genes is notable, and consistent with other disorders of the epigenetic machinery. The more dramatic changes in the R882 samples are interesting here as well, pointing to a more severe effect as the authors suggest; but again, this impact is blunted by questions about the study design.

RNA-sequencing and cell type:

The scRNA-seq work provides valuable insights into the changing cell populations in this disorder; however, as it was performed on a single DOS case and his healthy brother, it is difficult to discern which reported DEG can be attributed specifically to disorder status. As well, DEGs identified from isolated cell types were compared to DMRs identified from whole blood in additional samples. As such, this comparison generates many caveats, namely that DMRs may be a results of cell composition differences between groups (also reported in this paper) and that comparing cell-type differences between technologies may overwhelm any overlapping findings between expression and methylation. Authors also performed bulk RNA-seq on 4 healthy donors and 2 DOS patients but do not report a genome-wide analysis. These data may be more directly comparable with the methylation findings.

Also, in scRNA-seq data, it looks like the CD3+ T-cells have more distinct groupings in the case

than in the control - can the authors comment on this? Can they also try to identify stem cells? There may be important differences there. HSCs are present in peripheral blood - just in low numbers so it would depend on the total number of cells they sequenced.

Mouse work:

The genetic homogeneity of inbred mouse strains permits the use of this sample size in the WGBS mouse experiment. However, it is not clear if males and female were used nor if sex was used as a covariate. It appears age was not, even though there was an age range in the samples. Again, this should be accounted for in the statistical model. If both sexes were used this must be done. Again, as in the human WGBS, no details are given on the statistical parameters used to generate the DMRs. The paper that is cited to provide more methods details (Cole et al) does not have these details either. Parameters such a covariates are needed to interpret the study. The authors are appropriately cautious not to over interpret the lack of similarity between mouse and human DNAm data, still some quantitative data on the number of genes that overlap between the analyses would be welcome.

The mouse biochemical and behavioural work appears to have been performed and reported rigorously, the findings are notable and relevant to TBRS. The only note of caution is to consider correcting for multiple testing when reporting p-values from different parameters from the same test, though this is not settled as standard practice in the literature. If these changes can be made to the model, the mouse WGBS data would be a completing story; the changes in the -/- vs +/- and +/- bone marrow is very interesting and helps explain the phenotypic changes noted. The lack of correlation between gene expression and DNAm across the experiments is not surprising, very few studies find such a relationship. The experiment on Hoxb4 expression supports a specific role for DNAm in the expression of this gene. However, further validation could come from correlating the individual DNAm level at Hoxb4 and its expression level in each replicate animal to show this relationship.

Response to Reviews

The main changes in the revision include the following:

1. We added 9 additional healthy donor peripheral blood samples from individuals aged 4-17 years to better age- and sex-match the samples from the 11 DOS patients. These data did not change our overall conclusions, but did strengthen them.
2. With the additional DMRs controls, we performed covariate analyses to evaluate the roles of age and sex on the DMRs caused by DNMT3A mutations. Neither was a significant covariant.
3. We reevaluated all human and mouse scRNA-seq data with unbiased graph-based clustering, identifying the stages and lineages of each sample using two independent methods. Population shifts and DEG analyses were completely redone using these approaches. While some new features of the populations were better defined, the overall conclusions were not altered with this reanalysis.
4. All of the figures were reorganized to reduce crowding, increase font size, and improve the reader experience.
5. Additional references were added as suggested.
6. Language that was suggested to be misleading was corrected.

Responses to all of the specific comments from each reviewer are shown in **green** below.

Reviewer #1 (Remarks to the Author):

Smith and colleagues explore the consequences of DNMT3A mutations in patients with DOS syndrome and present a mouse model for one mutation found in DOS. The overall conclusions are that the R882 mutation presents a more severe mutation type than others which is manifest in a greater loss of DNA methylation. A great deal of ancillary data supports these overall conclusions.

While the data are of interest to specialists in the field, the figures are over-packed and poorly described, making extraction of the messages very difficult. At a minimum, the figures and data presentation need an overhaul so that the work is more accessible to the audience of Nature Communications readers. Also, the authors also must perform some actual validations of the data they present to support the claims. Many concerns are cited below. Many are concerned with data presentation that will make it easier for the reader to view and understand the data.

Concerns (no particular order of importance):

1. The authors claim on page 3 that R882 is “the most common” site of missense mutation in DOS. While technically true, given that only about 100 patients world-wide are published, and as yet, only a very small number have R882, it seems a bit of an overstatement to claim that it is “common”.

We have changed the text to reflect the small numbers of patients studied to date. The sentence in question now reads, “While the number of patients described in the literature is relatively small, the most frequent mutations alter amino acid position R882, which is also the most common site of *DNMT3A* mutations in patients with AML.” We note elsewhere that of the 100 cases described in the literature, 12 have mutations at R882.

2. There are many cases of mis-citing of papers or instances where other papers should be

cited. Page 3 and 14, the authors cite Spencer et al in support of R882 being dominant negative, but Kim et al (Blood, 2013) were the first to show this and should also be cited. I don't think citation (32) supports the dominant-negative hypothesis. In addition, Several other papers previously showed the loss of DNA methylation promoters, enhancers, gene bodies etc in mouse and human samples- both reduced methylation and the enlargement of regions. These include Tovy (Cell Stem Cell, 2020- human cell line), Jeong (Nature Genetics, 2014, mouse, Jeong, Cell Reports). The authors suggest on page 10 they are the first to look at WGBS in the context of "deficiency", but such data have been around for years in other papers, albeit not from a germline model. At least this prior work could be cited, particularly given that the overall observations concerning DNA methylation changes are essentially the same.

On page 3 and 14, we cite reference #18 regarding the dominant negative effect in hematopoietic cells (Russler-Germain, Spencer et al. 2014); Kim et al (Kim, Zhao et al. 2013) first described the effect in ES cells, which is now cited as well. The references to Jeong et al, Tovy et al, and Kim et al. have been added to the introduction. The statement pertinent to reference 32 has been removed from the discussion on 14.

We made no claim (nor do we intend to imply) that we were the first to look at WGBS in the context of *Dnmt3a* deficiency. We simply note that we obtained concurrent data for this study to compare with the germline R878H samples, where WGBS data has not previously been reported.

3. On page 6, the authors note a difference between patient and sibling of NK and T cell numbers and comment on "a potential lineage switch within this compartment". The results section is no place for wild speculation based on one patient. Also, NK cells are not considered part of the same "compartment" with classic T-cells.

NK and T cells both arise from Common Lymphoid Progenitors (CLPs). There are well described transcriptional (MYC), cell surface (CD27, CD244) and signaling (IL-15, IL7Ra) factors that contribute to the decision of a CLP to become a T-cell or NK-cell precursor, such as p-NKP or rNKP (Rosmaraki, Douagi et al. 2001, Fathman, Bhattacharya et al. 2011). Therefore, it may be possible that these lineages could be altered by an epigenetic change that alters terminal lineage commitment. However, since we have no direct evidence to support this hypothesis, we therefore removed it from the Results.

4. Page 9 refers to figure 5, but actually I believe this should be referring to figure 4.

We have corrected this error on page 9.

5. The authors use single-cell sequencing extensively to look at mouse and human samples. However, the authors should be more circumspect when drawing conclusions about gene expression differences. For example, the authors make claims about gene expression of HoxB genes and Rasip1. The main figures clearly show that a larger number of cells have detectable transcripts in some samples, which the authors claim indicates the genes are expressed at higher or lower levels (the data shown don't support this). The supplement shows a graph with "LSmeans" on the Y-axis, but this term is not explained. It is presumably least-squares mean which can be used in some contexts to infer gene expression. To claim gene expression differences like this repeatedly and to justify amplifying the observation in the discussion, the authors should perform real-time PCR.

We have addressed this issue in the revised manuscript, providing both number of cells expressing the gene of interest, and the level of expression in cells with detectable transcripts (See Figure 6e). We make no claims regarding either gene's functional relevance for the DOS phenotype, and describe them as examples chosen for illustrative purposes. We directly state that "we do not suggest that dysregulation of these specific genes is directly relevant for the DOS phenotype" in the discussion on page 17, although both may have links to autism and other genetic syndromes.

Rather than perform qRT-PCR studies as a means of orthogonally validating the dysregulation of individual genes, we performed bulk RNA-seq of two R882H samples (including the same one used for the scRNA-seq analysis), which are shown in Supplementary Figure 2. These data provide a much broader look at DEGs than a qRT-PCR study, and the data corroborates the scRNA-seq data for many genes of interest. Independent datasets from several biological replicates for the mouse studies confirm the dysregulation of the example genes (and many others) in the hematopoietic cells of another species with the same mutation. Finally, we also demonstrate the same effect on HoxB gene regulation in Supplementary Figure 6, where we used scRNA-seq data from Dnmt3a deficient mice described in Ketkar et al, (Ketkar, Verdoni et al. 2020). Remethylation of DMRs in the HoxB gene cluster with DNMT3A addback restored Hoxb gene expression. Taken together, these results provide substantial orthogonal validation of the human scRNA-seq data for these genes.

Figures:

Most main figures suffer from severe overcrowding and placement in the PDF that resulted in their being cut off on the right and sometimes the top. Many of the panels have text that is far too small, or features that are far too small. While readers can blow up images online, many readers still print and read on paper, so the presentation should be legible without a magnifying glass. Some examples noted below.

The figures have been reorganized to be more reader friendly. We have removed panels to the supplement, allowing us to increase text size and reduce crowding (see revisions to Figures 1, 2, 3, 5, and 6).

Also, in general the supplemental information has negative data. I suggest eliminating panels that truly show no differences, and moving some of the information from the main figures into the supplement.

We do present some findings that are not statistically different in the supplement for completeness and transparency, particularly for the behavioral analyses. All of the behavioral testing data must be viewed as a whole for the proper interpretation of the phenotype of these mice. We therefore moved appropriate panels to the supplementary appendix as an important part of the datasets.

1. Figure 1- change color scheme in (a) to differ from that in (b) (confusing).

We have altered the domain boxes to be different to the color scheme for the mutation grouping.

2. 1c- presentation of circles makes it difficult to distinguish the groups and the significance below could be lined up.

We have cleaned the dataset to include pertinent annotated regions, removed the significance table, and added them to the figure to make it more streamlined and interpretable.

3. 1d- impossible to read the Y-axis

Text size has been increased.

4. 1f and g- somewhat redundant – move one to supplement.

We feel that these analyses are both essential to demonstrate a key finding of the work. Going into this study, it was far from clear whether R882 and non-R882 mutations in DNMT3A caused the same or different methylation consequences, or whether non-R882 mutations would even have the same methylation phenotype--or unique ones that were mutation-specific. By performing independent DMR calling for the R882H and non-R882 mutant samples, we were able to demonstrate that these mutations affect methylation in the same regions of the genome, and show that the R882 mutation has a more severe hypomethylation phenotype than non-R882 mutations. Mechanistic studies of non-R882 mutations underway in several labs, and this data represents an important foundation for understanding the consequences of non-R882 mutations.

5. 1h/l are not well labelled. What are the green bars? Not indicated in legend. Difficult to discern the claimed DNA methylation difference in the hoxB cluster.

We have also used a different presentation method in the revised figures, and have more clearly highlighted the relevant DMRs and the data tracks.

6. Figure 2- most panels the fonts are far too small. (c) very difficult to read gene names.(f) contains a black bar to the left which appears to be gene clusters. The red squares are difficult to glean information from. Only informative panel (that is readable) is (h). Is expression detected at all in the controls?

We have reformatted this entire figure to incorporate changes recommended by reviewers 1 and 2.

7. Figure 4- (k) is not listed in the legend and I guess it is redundant with (b).

We have removed panel k and rearranged the figure to make it more reader friendly.

8. Figure 6- (b) 2 colors of blue look the same. (c) many panels are uninformative- could show the ones that are relevant and move to supplement or make a table. Not much to see in these panels except the number of genes. (d) is not well explained at all. There are green boxes and blue boxes. They refer to “violin” plots which are nowhere to be found. Be extreme scrutiny, it seems that the main feature of the graph is a few outliers at one extreme of violin plots that have been smashed against the origin. The graph does not show expression differences, but cells in which a signal for that gene is detected, not to be confused with expression level.

We have changed this figure as suggested to be more informative and reader friendly, and to clearly delineate the fraction of cells expressing the genes of interest, and the level of expression within positive cells.

9. Figure 5- cant tell the difference between pink and red on the plots. And for all the panels, the presentation could be better. The entire figure does not show much new.

We have changed the values from the $Dnmt3a^{+/-}$ mice to be represented in black.

There are no previous papers that have used WGBS to compare the methylation phenotypes of the hematopoietic cells of mice with germline mutations of $Dnmt3a^{R878H/+}$, $Dnmt3a^{-/-}$ and $Dnmt3a^{+/-}$. These data are important for understanding the relative methylation consequences for complete loss of function, heterozygous loss of function, and the R878H mutation. Since these findings have not previously been reported, and since they are fundamental for understanding the nature of the methylation consequences of the R882/R878 mutations, we feel that they are significant and merit reporting.

10. Supp fig 1- suffers from the same presentation problems as Fig 1. The text states there are no differences in global methylation at the indicated features, but there are p-values given which appear to be significant. Which is it? Ditto figure Supp 6 except the null does appear to have some lower methylation even at a global level.

In response to requests from other reviewers, we added WGBS data from 9 more healthy donor peripheral blood samples from males and females that are better age-matched to the DOS cohort. With the additional control samples, the global methylation value for $DNMT3A^{non-R882}$ now reaches a p-value 0.0079, which is reflected in the revised figures and text. P values are listed for all possible comparisons in the revised figures rather than in table format.

11. Figure 8. (c) shows leukemia free survival. The WT mice show no apparent deaths out to 800 days. In most facilities, at least $\frac{1}{2}$ of WT animals would be lost by this time point. Did the authors really have no deaths? Or were they just not leukemia deaths? Did they actually examine the cause of death of the WT and ascertain they were leukemia free? I buy the overall conclusions, but probably this could be presented differently to account for deaths of animals that were not able to be determined.

This figure indeed represents leukemia free survival. There were no deaths directly attributable to leukemia in the WT cohort. We necropsied many moribund WT mice at advanced ages and found many cases of lymphoma, a known cause of mortality in B16/J mice, and other deaths without obvious causes. We censored these mice, and any others whose cause of death was not caused by overt leukemia.

12. Figure 7- way way too much data to look at on the right with the small plots, and the text is illegible. What is this trying to show? What is new information here?

We have modified this figure, and since it is primarily corroborative, moved it to the supplement. The dataset is important for fully understanding the model, since it provides a detailed look at hematopoietic populations in germline R878H mice.

13. Supp Fig 7- confusing and LSmeans is not strictly quantification of gene expression. What are B and D trying to show?

We have changed the figure so that the Y axis represents the measurement of expression values. B and D show expression values in specific subsets of cells.

14. All figures should be carefully re-examined for legibility and messaging.

We have done so, and where ever possible, made them simpler and more reader friendly, hopefully without compromising content.

Reviewer #2 (Remarks to the Author):

In the manuscript, Smith and colleagues correlate functional mutations in DNMT3A with DNMT3A Overgrowth Syndrome (DOS) and hematopoietic malignancies. The authors whole genome bisulfite sequencing on peripheral blood of DOS patients harboring the DNMT3A R882 (n=3) and DNMT3A non-R882 (n=8). This analysis identified subtle decreases in methylation at CPG islands and promoters, but not gene bodies or enhancers for both the R882 and non R882 mutants. In addition, the R882 mutants showed significantly more differentially methylated regions (DMRs) than non-R882 when compared to the DNMT3A+/+ samples, suggesting that the methylation phenotype is more severe in the mutant patients.

Next, the authors generated a germline heterozygous Dnmt3a R878H/+ mouse model to investigate the molecular mechanisms promoting the pathogenesis. The Dnmt3a R878H/+ mutant recapitulated phenotypic symptoms observed in humans such as obesity, longer femur and humerus lengths, and macrocephaly associated to behavioral issues such as volitional movement deficits, and emotional and several cognitive impairment. WGBS in bone marrow cells collected from wild-type, Dnmt3a R878H/+ and Dnmt3a-/- mice demonstrated that an overlap in the DMR of Dnmt3a R878H/+ and Dnmt3a-/- samples. However, the average methylation difference was greatest in the ko cells. In addition, scRNAseq in whole bone marrow cells collected from Dnmt3a R878H/+ and WT counterparts young (1 month) and aged (9 months) mice showed enrichment for hematopoietic stem and progenitor cells (HSPCs) and monocytes in the old mutant mice. This analysis also showed downregulation of Hoxb4 in PMNs, GMPS, and monocytes from the Dnmt3a R878H/+ mutant mice compared to the wild-type. Finally, cytotoxic challenges using standard chemotherapy such as doxorubicin and cytarabine decrease in overall survival of the mutant mice.

This is an interesting study analyzing the methylation profile of hematopoietic cells from DOS patients. The authors presented a novel mouse model to investigate the mechanisms of DOS. The manuscript includes compelling evidence of the fidelity of the mouse model to replicate observations made in human patients. Overall, experiments are well designed and analyzed. However, the single cell RNA-seq data (Figure 6) is poorly presented. Moreover, the statistical analysis applied in the single-cell RNA-seq to compare the cell distribution in groups which contain only one sample (or is it two?) is unclear. The authors should refer to previous studies on mouse bone marrow samples (Giladi A., et al 2018, Izzo F et., 2020...) for a better analysis and representation of the scRNA-seq data. The molecular phenotypes described in the hematopoietic cells are interesting.

Major comments:

- Figure 2: The statistical analysis of the single cell RNA-seq from the peripheral blood of the patient samples is questionable. The authors analyze two samples of the same patient (one fresh and one frozen) and compare to one healthy donor. Therefore, it is only one patient per

group. Although an analysis of distribution of the populations per sample is interesting, applying an statistical analysis (as shown in 2d) in only one sample per group is not correct. Authors should show the distribution of the cells per sample in bar plots or in a pie representation rather than statistical analysis between groups containing only one sample.

We removed the cryopreserved sample from this analysis, because it was not as informative as the fresh samples from the patient and his brother, which were processed in the same way on the same day. Fresh samples could not be processed from the other 10 patients, since none were local, and since airline travel for these families during the COVID pandemic posed a significant barrier. Cryopreserved peripheral blood samples do not perform well in scRNA-seq studies, almost certainly because of the presence of large numbers of neutrophils, which release their contents upon thawing, causing fratricide and other deleterious effects. We now have one sample per group, and the percentage of different cell types is reflected appropriately in the figure as a bar plot. The same samples were evaluated with a 15 color flow panel, which corroborated the population shifts detected by scRNA-seq (Supp Figure 2c).

- Figure 2c: A better strategy for cell calling including several genes per group should be included in the main or supplementary figure. For instance a heatmap representation of genes in each population may be a better approach to highlight the gene profile of each cell type. How do the authors explain the lack of monocytes? Are some of the monocytes included in the neutrophil cluster?

We have reanalyzed this data using graph-based clustering and used ToppGene analyses to identify cell populations. We then validated these populations with a minimum of two known population markers, which is shown in the revised supplemental figure 2. This reanalysis resulted in the identification of 2 clusters that clearly identify as monocytes (Figure 2a, clusters 6 and 11).

Unfortunately, due to space and length constraints, we cannot include heatmaps for each graph-based cluster, which would create 12 heatmaps for the human samples and 24 for the mouse. However, we have included a heatmap for those genes that are differentially expressed in more than one cluster (Figure 2d), and we provide complete expression data of differentially expressed genes as datasets to be used by interested parties (Datasets 03, 04 and 08). We have also provided two specific examples of differentially expressed genes in the human and mouse with very granular representations of expression (reads per cell and percent of positive cells; Figure 2e).

- Figure 2e and f: Regardless of the GO mentioned on the NK cells, did the authors find upregulation or downregulation of specific pathways in other cell types? Is there any metabolic gene dysregulated? Or any other gene that could be extrapolated to the overgrowth syndrome?

We have now shown GO terms for all graph-based clusters containing differentially expressed genes (Supp Figure 2D). While there is enrichment for several pathways, the majority pertain to hematopoietic function. Angiogenesis and blood vessel and endothelial cell migration are enriched GO terms, but nothing strictly relating to overgrowth and metabolism were identified. This is not surprising to us, since the blood counts of non-leukemic DOS patients are normal. Further, extensive metabolism studies from the plasma of R878H mice (including glucose, FFA and leptin) were not overtly dysregulated (Supp Figure 3). Based on these findings, we suggest that it is unlikely that hematopoietic cells drive the overgrowth phenotype in these patients. The cause of the overgrowth phenotype will require additional studies to define it, which is mentioned in the discussion.

- Figure 3a, b, j, k: It seems that the differences in weight differences are more pronounced as the WT and mutant age. What is the effect of the mutation on metabolism? Is there an age dependent metabolic impact of the Dnmt3a R878H/+ mutant? Authors could analyze metabolic profiles in the serum of the mice.

The reviewer may not have noticed the extensive data on metabolic parameters from mouse plasma, shown in Supp figure 3. These are reemphasized in the revised manuscript.

- Figure 4: Did the authors collect brain H&E in samples from the mice? Does the brain of mutant mice show major structural abnormalities?

This is an interesting and important point, but we respectfully submit that this is beyond the scope of this study. A complete neuropathological analysis of these mice is now justified, since we have proved that these mice have a behavioral phenotype, and a collaboration with a qualified neuropathologist is in progress.

- Figure 5f & g: Why are there no Dnmt3a^{+/+} 52 weeks of age samples while there are samples for the Dnmt3a ^{+/-} mutants? A heatmap or barplot for mean methylation values comparing age-matched or cohorts with a similar age group would be more clear and easier to interpret than how the data is presented currently. Since the mutation seems to have some age dependent effect, separating the cohorts might be more clear as well.

We have added a 52-week old control sample in this dataset. In additional analyses of these data, we have determined that there is no age-dependent evolution of the DMR phenotype in hematopoietic cells in the R878H samples. We have also performed covariate analysis for age and sex, and found there are no DMRs in the R878H cohort that are dependent on either age or sex.

- Figure 6a: an UMAP combining all the samples together should be included in the figure, together with the independent ones.

This is now included in the supplement for this figure (Supplemental Figure 5a).

- Figure 6a: Did the authors perform unbiased clustering? Were the samples normalized to the number of cells for each condition? Do the mutant samples cluster more closely together? Do the mutants show the emergence of any mutant specific cluster in the different populations? A density plot might show differences in transcriptional expression between the different conditions.

We have performed unbiased graph-based clustering, and have now included this analysis in the mouse scRNA-seq figure (Figure 6 and Supp Figure 5). The samples were normalized for cell numbers and all other QC metrics. The mutant samples essentially overlay the WT samples. The populations of each cluster for each sample are highlighted in a pie chart, as suggested, which shows that for the majority of clusters, there are small age-dependent shifts, rather than genotype-related shifts. There is one very small cluster that is expanded in the 9 month old R878H sample, which is now noted; additional work will be required to better understand this cluster, which we have only seen in one sample to date. With a large number of graph-based clusters and four samples in the study, detailed analysis of gene expression differences in individual clusters would be impossible to present considering figure and length

constraints from the journal. The entire database will be available in the SRA for more detailed analyses by interested parties.

- What are the highly expressed genes for each cell population? Cell-specific heatmaps should be included in this figure or supplementary.

As noted above, this would require 24 additional tables and/or heatmaps. We are constrained in the number of supplementary figures that we can present in Nature Communications, and suggest that the deposition of these data in the SRA will allow other investigators to define these changes in specific populations.

- Figure 6b: statistical analysis between groups that include only one sample is dubious at best. How many samples and individual mice were actually used in Figure 6? At least 3 animals in each timepoint should be used.

We evaluated one mouse each at two separate times (1 vs. 9 months) for this study. The cost of performing scRNA-seq in triplicate for each time point is simply beyond the budget of this lab: the data production costs alone would be in excess of \$35,000. Furthermore, the two samples from each timepoint are remarkably similar to each other in terms of cellular populations (see Figure 6a), which is the key point of this paper. These data were corroborated by 21 color flow analysis of bone marrow and peripheral blood cells for 17-31 individual mice of different ages from each genotype (Supp Figure 7). Finally, scRNA-seq data is different from bulk RNA data in terms of statistical analysis, since data from thousands of individual cellular 'experiments' are available within each sample. Few if any published studies of this kind have performed scRNA-seq experiments in triplicate for multiple time points. If this is indeed the new standard for these studies, we will no longer be able to afford them.

- Figure 6: What were the phenotypes of the mice when they were used for the experiment? What was the disease severity of the mice?

As noted in additional figures in the study, the growth phenotypes were very subtle at 1 month of age, and became more pronounced at age 9 months. Behavioral phenotypes were not assessed at multiple time points, but the data were obtained over a 100-day time period in young mice. DMRs defined by the R878H mutation did not change significantly as a function of age, as noted above.

- Figure 8b: Do the authors have an explanation for the variability in the transplantation experiment? Was there an observed difference in the mice that expanded in the host vs the one that did not engraft?

This experiment has been performed in 'somatic' R878H mice by others. Because of the small sample size, we decided to remove this figure from the paper, since it was not relevant for the major conclusions of the study.

- Error bars and/or Statistical analysis should be included in the graphs of the supplementary figures: 6, 7 and 8.

Supp Figure 2, 3, 4, 5, 6 and 7 do have statistical analyses included in the revised figures.

Supp 5- no error bars, since n=1

Supp 7- no error bars, since n=1

For Supp Figure 6, the global methylation data has been moved to the main figure 5, and stats are defined directly on the graph.

Supp 5-the statistical analysis done here was a Fisher's Exact test.

Minor comments:

- Introduction: What is the percentage of patients with DOS that harbor the DNMT3A R882 mutations? How many of these patients develop hematologic malignancies and is it the leading cause of mortality within DOS patients?

Since DOS is a recently described syndrome, individuals are only now being identified and natural history studies have not yet been published. Of the 100 patients described in the literature, 12 have mutations at R882. Only 200 patients have been identified world-wide, but most have not been described in publications. In the discussion, we describe what is currently known about leukemia risk in DOS patients, which we are now formalizing in a clinical report that will be submitted elsewhere.

- Mice with germline Dnmt3a R878H/+ exhibit behavioral abnormalities: Text does not match Fig 5 a-k, but refer to Figure 4 and Supplementary Fig 5 a-n.

This has been corrected in the revision.

- Fig 8b Do each of the lines represent one sample or a cohort of samples? Please clarify.

As noted above, we have decided to remove this figure from the manuscript.

Reviewer #3 (Remarks to the Author):

This is a very interesting study describing the DNA methylation phenotypes and its transcriptional effects in humans and mice with DNMT3A overgrowth syndrome (DOS). The data presented suggest that many DNMT3A mutations associated with DOS decrease the methyltransferase activity of DNMT3A, causing a focal, canonical hypomethylation phenotype that is most pronounced in DNMT3A R882 mutations, most commonly found in cancer.

Major comments

DNA methylation analysis:

The authors used n=11 DOS (or TBRS) cases ranging from 20 months to 36 years compared to n=6 young adult controls. The first concern with this study design is the sample size. N=11 TBRS cases should be sufficient to identify DNA methylation changes across the genome with large effect sizes. However, cases were separated into subsequent groups based on the presence or absence of an R882 variant (n=3 with R882 and n=8 without) and each group was compared to just 6 controls. This number of controls is smaller than what is optimal to identify DNA methylation signatures. Currently, 2-3 times the number of cases is the target for the controls.

We have added 9 additional control WGBS samples from healthy donors ranging from 4-17 years of age. However, we feel that the request for WGBS data from 33 normal controls is extreme, especially since the global methylation phenotype of blood cells is so similar among healthy individuals in this study, and many others. While it would be wonderful to have a resource of hundreds of normal peripheral blood samples of different ages and sexes for this comparison,

the request for this study represents a new standard that we cannot address in a timely fashion. Further, we are quite certain, based on the minimal variation among the healthy donors, that adding another 16 control samples to the study will not change the results reported here.

The age range of the controls, as compared to the cases, needs to be matched given the strong association between age and DNAm (especially with such a wide age range within the cases), i.e. the control age range should reflect the 20 month-36 year range of the cases. As well, in silico work-arounds such as adjustments for blood cell composition (estimated via DNA methylation) are not yet available for WGBS data, as they are for microarray data and this should be presented as a limitation of the study. Additionally, it is not clear that sex was taken into account for DMR identification (i.e. used as a covariate), which is necessary.

We have added 9 additional control samples ranging in age from 4-17 years of age, from both sexes, as shown in Figure 1. While blood cell composition is an important consideration, the blood counts of non-leukemic DOS patients are essentially normal—as they are for thousands of people with DNMT3A mutations and clonal hematopoiesis (the definition of clonal hematopoiesis requires normal blood counts). While we study only one patient here for the logistical reasons noted above, there have also been other studies from mice showing that the methylation phenotypes established by *Dnmt3a* deficiency across cell types (from progenitors to mature lineages) is very consistent (Ketkar, Verdoni et al. 2020). Furthermore, WGBS is different from array-based methylation methodologies because the entire CpG space in the genome is covered without bias.

We have performed age and sex covariate analyses for both the human and mouse datasets, and found that neither variable contributed to the *Dnmt3a* dependent DMRs identified in this study. This is now stated in the manuscript.

The authors state that “To date, the methylation phenotypes of DOS patients with non R882 mutations have not yet been described.”. However there have been at least 2 papers that report DNA methylation profiles for non R882 variants as well as other variants -see Jeffries et al 2019, PMID: 31160375 and Aref-Eshghi et al 2020, PMID: 32109418.

Both of the described studies used array-based platforms to describe the methylation profiles in these patients. We have now changed this sentence to read “To date, the global methylation phenotypes (defined by WGBS) of DOS patients with non-R882 mutations have not yet been described and compared to R882 mutations, although array-based methylation studies of the blood cells of these patients have been reported (Jeffries, Maroofian et al. 2019, Aref-Eshghi, Kerkhof et al. 2020).”

In addition, it is important to note that Jeffries et al PMID: 31160375 have reported age acceleration in DNMT3A variants and that age acceleration is much higher with R882 mutations. Have the authors looked into the effect of DNA methylation aging in their cohort? Maybe using DNA methylation aging as a confounding variable can further support their results.

As described in the Methods section, DMRs were first called using metilene and then filtered to restrict to only ‘canonical’ regions with low variability within each comparator group (in particular, we required $sd < 0.1$ in both ‘control’ and ‘mutant’ groups for each comparison). Thus, regions strongly affected by age and/or sex differences in methylation would have been excluded from our analysis. In addition, we now have performed linear regression with $\log(\text{age})$ and sex

included as covariates. **Out of 24,882 total DMRs identified in this study for human and mouse, only one (identified in Dnmt3a^{-/-}) lost statistical significance (FDR<0.05) after adjustment for sex and age.** For the human patients, only one sample was obtained at each age, and most of the mutations were unique. For the mouse samples, multiple ages were examined, but the DMR phenotype did not significantly progress as a function of age.

Figure 1a: It is highly recommended to add the mutation information for all variants in this figure and replace the sample numbers.

This has been done.

There is no detailed information about the mutations i.e. cDNA and protein information. Also, there is no information about the clinical phenotype of the 11 DOS patients. The authors refer to clinical information in table 1 but table 1 is missing from the manuscript.

We have added the mutational information to the heatmaps in Figure 1. Table 1 was inadvertently omitted in the original manuscript, but it is now provided.

The statement “Of these DMRs, 134 (72%) overlapped with DMRs identified in the DNMT3A R882 samples (where overlap required at least 1bp of shared sequence), “ suggests the fact that these 2 identified DNA methylation phenotypes correspond to one DNMT3A DNA methylation profile and the 28% that did not overlap between the 2 could be the result of power issues/sample size difference between the 2 groups. Have the authors considered generating a DNA methylation profile for all DNMT3A variants?

We have now updated these data and compared them to the additional controls, which has increased the numbers of DMRs detected. We have indeed considered the power/size issues, but are constrained by the rarity of the disease and logistical issues noted repeatedly above. Despite these constraints affecting the analysis of individual DMRs, the data clearly show that the methylation phenotype of the R882 patients is more severe than the non-R882 patients, and that the DMRs are occurring in essentially the same genomic locations. These data suggest that the mechanisms underlying each non-R882 mutation merit individual investigation, which is now ongoing in several labs. These data from primary human samples create an important foundation for genotype-phenotype correlations.

“The fraction of T cells was reduced (46.04% to 11.30.29%, $p = 3.017e-13$), and the NK cell fraction was increased (3.37% to 14.88%, $p \leq 12.2e-16$), suggesting a potential lineage switch within this compartment.” The authors state that this analysis was done comparing the patient to his unaffected sibling who is 4 year older than the patient. Unfortunately, the authors did not provide the age of the affected sibling. It is important to know the age to better understand the shift in cell types reported here and if it is due solely to age difference rather than DNMT3A status. (The patients included in this study are as young as 20 months old).

The ages are now available in Table 1 (DOS1; 9y, Sibling; 13y). This was an interesting observation that was validated by flow cytometry, However, since this event was identified in one patient at one timepoint, we have dropped the possible explanation of this finding since it is not supportable with additional information.

Given the observed difference in the phenotype with aging in mice with a germline DNMT3A heterozygous variant, it is recommended that the authors discuss what is known about the

clinical features of DOS in adult patients compared to children. How do the clinical features of their older patients compare to the younger ones?

To date, there has not yet been a study describing the natural history of DOS patients, since the syndrome was described only 7 years ago, and since it is so rare. No clear genotype:phenotype correlations have yet been established in the literature. We hope that the data included in this paper begins to frame some of these questions for future studies.

Minor Comments:

The authors state “To determine whether DOS patients have DNA methylation changes in the genomes of their hematopoietic cells, we obtained peripheral blood and buccal swab samples from children and adults with DOS”. It is not clear if buccal samples were used for DNA methylation analyses as well.

The buccal swab DNA samples were used only to validate the germline origin of DNMT3A mutation. They were also compared to the blood samples to identify any potential cooperating mutations that might contribute to subtle clonal expansions in blood cells (none of the patients had evidence for the latter). We have now changed this sentence to make it clear that only the peripheral blood samples were used for methylation studies.

Figure 1h and 1i can be supplemented with violin plots or boxplots to better represent the DNA methylation differences at the selected DMRs between the 3 groups.

We have replaced 1h and 1i with a more summarized format.

With such small numbers and an unbalanced study design, it is difficult to know if any other reported differences between the different variant types are meaningful. The effect of the R882 variant does indeed appear dramatic on the heat map (1F); however, it is difficult to discern the effect of age and stochastic changes in the small sample size from those of the disorder. This issue extends to comparing the DMR severity and size features of R882 variants vs. non-R882 variants.

Table 1 seems to be missing so can't assess sex, age, and clinical features in the R882 variants vs. non-R882 variants.

As noted above, Table 1 is now included.

Age and sex were investigated as contributing covariates and were found to contribute minimally to the DNMT3A-dependent DMRs identified.

The sample size is small because of the rarity of these patients, because the children with DOS have very problematic behavioral issues that make travel, physician visits, and invasive procedures extremely difficult (or impossible) for some of the children. We have worked with the only family support group in the US to identify and study as many patients as possible for this report. Adding more patients to the study at this time would be nearly impossible to achieve, and it is very unlikely that additional patients will change the major findings presented here.

Dismissing the canonical DMR findings for the R882 patients because of the small sample size is very difficult for us to understand, since the findings are so reproducible among different patients, and since they are confirmed in the mouse model of the R882H mutation.

In general, because of the study design, more care should be taken in drawing conclusions, In

the discussion, the authors state: “Since all of the 6 other non-R882 missense mutations had a remarkably similar methylation profile 7 (Figure 1g), we suggest that all non-R882 mutations essentially inactivated the affected allele”. The similarity of DNAm profiles cannot yet be used to make these kinds of determinations, even when statistical methods are more rigorous than those used here. This represents an over-interpretation of the data.

The identification of changes in genes such as HOX genes is notable, and consistent with other disorders of the epigenetic machinery. The more dramatic changes in the R882 samples are interesting here as well, pointing to a more severe effect as the authors suggest; but again, this impact is blunted by questions about the study design.

We have changed this sentence to read “Since all of the other non-R882 missense mutations had a remarkably similar magnitude of methylation loss (Figure 1), we suggest that the non-R882 mutations may inactivate the affected allele, although the mechanisms of inactivation are almost certainly multifarious.”

As the reviewer suggests, a perfect study would include multiple samples from a worldwide cohort of DOS patients with sex and age-similar unaffected siblings. We do not have the resources to perform such a study. Until this can be done, smaller studies that evaluate as many DOS patients as possible will provide important insights that can still have an impact on our understanding of this syndrome.

RNA-sequencing and cell type:

The scRNA-seq work provides valuable insights into the changing cell populations in this disorder; however, as it was performed on a single DOS case and his healthy brother, it is difficult to discern which reported DEG can be attributed specifically to disorder status. As well, DEGs identified from isolated cell types were compared to DMRs identified from whole blood in additional samples. As such, this comparison generates many caveats, namely that DMRs may be a results of cell composition differences between groups (also reported in this paper) and that comparing cell-type differences between technologies may overwhelm any overlapping findings between expression and methylation. Authors also performed bulk RNA-seq on 4 healthy donors and 2 DOS patients but do not report a genome-wide analysis. These data may be more directly comparable with the methylation findings.

We do appreciate that having one peripheral blood sample pair for DOS patient and his healthy brother is less than optimal. However, obtaining more fresh samples is logistically nearly impossible at this time, for the many reasons cited above. We have done our best to place the limited findings in proper context. Because samples from these rare patients are so difficult to obtain, we felt that providing even one example of high-quality data is better than nothing. However, if the reviewer insists that the data is inadequate and the editors agree, the data could potentially be removed from the paper. In addition, the validation of genes that are differentially expressed with bulk RNA-seq (including an additional R882H patient) suggests there are indeed canonically dysregulated genes. All of the primary data from these studies has been deposited to dbGaP, and spreadsheets defining the DEGs are in the Supplemental materials (Datasets 03, 04 and 08). Further, the comparison of expression data across species with the same Dnmt3a mutation provides another form of orthogonality that should not be dismissed.

Also, in scRNA-seq data, it looks like the CD3+ T-cells have more distinct groupings in the case than in the control - can the authors comment on this? Can they also try to identify stem cells?

HSCs are very rare in peripheral blood and were not detected in the study of these two individuals. Obtaining bone marrow samples from these individuals to increase the yield of

potential HSCs is a study that we would like to do, but it would require general anesthesia for this DOS patient, which is not ethically justified for this research study.

We performed unbiased graph-based clustering followed by ToppGene analysis to identify cellular populations based on gene expression. This has allowed us reclassify the T-cell clusters, as shown in the revised Figure 2a and 2b.

Mouse work:

The genetic homogeneity of inbred mouse strains permits the use of this sample size in the WGBS mouse experiment. However, it is not clear if males and female were used nor if sex was used as a covariate. It appears age was not, even though there was an age range in the samples. Again, this should be accounted for in the statistical model. If both sexes were used this must be done. Again, as in the human WGBS, no details are given on the statistical parameters used to generate the DMRs. The paper that is cited to provide more methods details (Cole et al) does not have these details either. Parameters such a covariates are needed to interpret the study. The authors are appropriately cautious not to over interpret the lack of similarity between mouse and human DNAm data, still some quantitative data on the number of genes that overlap between the analyses would be welcome.

As mentioned above, we did age and sex covariate analysis, and found that these did not significantly contribute to the Dnmt3a-defined DMRs.

We have updated the methods to include specific information with regard to WGBS analysis. It now reads "The 'metilene' algorithm was used to analyze the methylation ratios at all CpGs to identify differentially methylated regions (DMRs). Each DMR was required to span >10 CpGs, have a mean methylation difference between groups of >0.2, and a false discovery rate (FDR) <0.05. Adjacent DMRs <50 bp apart were merged, then "non-canonical" DMRs with standard deviation within either group of >0.1 were removed (Juhling, Kretzmer et al. 2016)."

In the revised manuscript, we have defined R882-associated DMRs that are shared between the human peripheral blood cells and the mouse bone marrow cells, as described on page 12.

The mouse biochemical and behavioural work appears to have been performed and reported rigorously, the findings are notable and relevant to TBRS. The only note of caution is to consider correcting for multiple testing when reporting p-values from different parameters from the same test, though this is not settled as standard practice in the literature. If these changes can be made to the model, the mouse WGBS data would be a completing story; the changes in the -/- vs +/- and +/- bone marrow is very interesting and helps explain the phenotypic changes noted.

We analyze the behavioral data with two-way repeated-measures ANOVA with Sidak's multiple comparison test where appropriate, as described in other publications presenting similar data (Christian, Wu et al. 2020)

The lack of correlation between gene expression and DNAm across the experiments is not surprising, very few studies find such a relationship. The experiment on Hoxb4 expression supports a specific role for DNAm in the expression of this gene. However, further validation could come from correlating the individual DNAm level at Hoxb4 and its expression level in each replicate animal to show this relationship.

Please note extensive comments regarding the orthogonal validation of Hoxb gene dysregulation, in response to Reviewer 2.

Aref-Eshghi, E., J. Kerkhof, V. P. Pedro, D. I. F. Groupe, M. Barat-Houari, N. Ruiz-Pallares, J. C. Andrau, D. Lacombe, J. Van-Gils, P. Fergelot, C. Dubourg, V. Cormier-Daire, S. Rondeau, F. Lecoquierre, P. Saugier-veber, G. Nicolas, G. Lesca, N. Chatron, D. Sanlaville, A. Vitobello, L. Faivre, C. Thauvin-Robinet, F. Laumonier, M. Raynaud, M. Alders, M. Mannens, P. Henneman, R. C. Hennekam, G. Velasco, C. Francastel, D. Ulveling, A. Ciolfi, S. Pizzi, M. Tartaglia, S. Heide, D. Heron, C. Mignot, B. Keren, S. Whalen, A. Afenjar, T. Bienvenu, P. M. Campeau, J. Rousseau, M. A. Levy, L. Brick, M. Kozenko, T. B. Balci, V. M. Siu, A. Stuart, M. Kadour, J. Masters, K. Takano, T. Kleefstra, N. de Leeuw, M. Field, M. Shaw, J. Gecz, P. J. Ainsworth, H. Lin, D. I. Rodenhiser, M. J. Friez, M. Tedder, J. A. Lee, B. R. DuPont, R. E. Stevenson, S. A. Skinner, C. E. Schwartz, D. Genevieve and B. Sadikovic (2020). "Evaluation of DNA Methylation Episignatures for Diagnosis and Phenotype Correlations in 42 Mendelian Neurodevelopmental Disorders." Am J Hum Genet **106**(3): 356-370.

Christian, D. L., D. Y. Wu, J. R. Martin, J. R. Moore, Y. R. Liu, A. W. Clemens, S. A. Nettles, N. M. Kirkland, T. Papouin, C. A. Hill, D. F. Wozniak, J. D. Dougherty and H. W. Gabel (2020). "DNMT3A Haploinsufficiency Results in Behavioral Deficits and Global Epigenomic Dysregulation Shared across Neurodevelopmental Disorders." Cell Rep **33**(8): 108416.

Fathman, J. W., D. Bhattacharya, M. A. Inlay, J. Seita, H. Karsunky and I. L. Weissman (2011). "Identification of the earliest natural killer cell-committed progenitor in murine bone marrow." Blood **118**(20): 5439-5447.

Jeffries, A. R., R. Maroofian, C. G. Salter, B. A. Chioza, H. E. Cross, M. A. Patton, E. Dempster, I. K. Temple, D. J. G. Mackay, F. I. Rezwan, L. Aksglaede, D. Baralle, T. Dabir, M. F. Hunter, A. Kamath, A. Kumar, R. Newbury-Ecob, A. Selicorni, A. Springer, L. Van Maldergem, V. Varghese, N. Yachelevich, K. Tatton-Brown, J. Mill, A. H. Crosby and E. L. Baple (2019). "Growth disrupting mutations in epigenetic regulatory molecules are associated with abnormalities of epigenetic aging." Genome Res **29**(7): 1057-1066.

Juhling, F., H. Kretzmer, S. H. Bernhart, C. Otto, P. F. Stadler and S. Hoffmann (2016). "metilene: fast and sensitive calling of differentially methylated regions from bisulfite sequencing data." Genome Res **26**(2): 256-262.

Ketkar, S., A. M. Verdoni, A. M. Smith, C. V. Bangert, E. R. Leight, D. Y. Chen, M. K. Brune, N. M. Helton, M. Hoock, D. R. George, C. Fronick, R. S. Fulton, S. M. Ramakrishnan, G. S. Chang, A. A. Petti, D. H. Spencer, C. A. Miller and T. J. Ley (2020). "Remethylation of Dnmt3a (-/-) hematopoietic cells is associated with partial correction of gene dysregulation and reduced myeloid skewing." Proc Natl Acad Sci U S A **117**(6): 3123-3134.

Kim, S. J., H. Zhao, S. Hardikar, A. K. Singh, M. A. Goodell and T. Chen (2013). "A DNMT3A mutation common in AML exhibits dominant-negative effects in murine ES cells." Blood **122**(25): 4086-4089.

Rosmaraki, E. E., I. Douagi, C. Roth, F. Colucci, A. Cumano and J. P. Di Santo (2001). "Identification of committed NK cell progenitors in adult murine bone marrow." Eur J Immunol **31**(6): 1900-1909.

Russler-Germain, D. A., D. H. Spencer, M. A. Young, T. L. Lamprecht, C. A. Miller, R. Fulton, M. R. Meyer, P. Erdmann-Gilmore, R. R. Townsend, R. K. Wilson and T. J. Ley (2014). "The R882H DNMT3A mutation associated with AML dominantly inhibits wild-type DNMT3A by blocking its ability to form active tetramers." Cancer Cell **25**(4): 442-454.

REVIEWER COMMENTS

Reviewer #2 (Remarks to the Author):

In the revised manuscript the authors provided clear answers to most of my comments. However, before publishing, we recommend that the authors address one concern regarding the analysis of the single RNA seq data. In principle, I agree with the authors that in single cell analysis one sample per group could be representative enough to achieve statistically significant differences since you compare a parameter measured in thousands of cells. For instance, when you compare the mean expression level of a gene in a cell population. In this case each group contains hundreds or thousands of cells ($n=100/1000\dots$ etc). However, when the authors analyzed the percentage of different cell types per group, for instance the percentage of B-cells in each group, they are measuring one parameter in a group which includes only one sample. Since $n=1$, the authors can't apply statistical analysis. Therefore, I suggest that the authors change the bar plots in Figure 2b and Supp 2b, by a pie chart as shown in Figure 6b.

Reviewer #3 (Remarks to the Author):

The authors have implemented many of the suggestion from all three reviewers, which have successfully improved the quality of the manuscript.

For the DNA methylation analysis, the authors have made efforts to implement many of the changes suggested. Particularly, the addition of 9 controls that better match the age distribution of the cases strengthens confidence that the effects seen are due to genotype and not age. I should clarify, that it was not the intention of this reviewer to imply that 33 controls should be used, indeed this is an unreasonable suggestion for this dataset. The suggestions was only meant to contextualize this study in the context of the standards that exist for DNAm microarray experiments to which this study will be compared. The effects of TBRS on DNAm relatively large compared to other disorders of the epigenetic machinery: the effect size is large and many CpG sties are affected, as data here and reported elsewhere show. Therefore, as the authors note, changes made to the experiment design and/or statistical analysis had minimal effects on the data output. Importantly, this is not a reason not to undertake proper statistical and experiment design approaches. WGBS data in epigenetic regulatory disorders are only now becoming common, and this paper will be part of setting these standards. This has been the reason for this reviewer's insistence on certain standards, even if the authors (rightly) feel they may have limited impact on the final data reported.

Specifically, the main issues are age and sex. The authors state that they have accounted for age and sex as covariates, and that this didn't impact the results. Again, this is not unexpected given the effect size. It is noted in the manuscript now that this was done, which is important. However, it is not clear if the data shown throughout are the result of this analysis. If all the DMRs remained the same, why not report the analysis with age and sex as covariates? The same is true for the mouse analysis. It appears that rather than report the new analysis, the original dataset was simply check against the new one with proper covariates used.

The author's response regarding cell composition is sufficient given the limitations of WGBS at this time. Noting that cell types are not reported to change is the best that can be done currently. However, the author's response that: "Furthermore, WGBS is different from array-based methylation methodologies because the entire CpG space in the genome is covered without bias." is confusing, it is unclear what relevance the difference between DNAm microarray and WGBS technology has here. This issue is whether differences in DNAm between case groups (detected using either method) are in part due to differences in changing cell populations in the case group, not due to actual differences in DNAm in those cells. Again, this cannot be addressed yet in WGBS data, as such it should be listed as a limitation.

The authors efforts to scale back the language around interpretation around of the difference in DNAm severity between the non-R882 and R882 variants is appreciated. The authors respond, "As the reviewer suggests, a perfect study would include multiple samples from a worldwide cohort of DOS patients with sex and age-similar unaffected siblings. We do not have the resources to perform such a study. Until this can be done, smaller studies that evaluate as many DOS patients as possible will provide important insights that can still have an impact on our understanding of this syndrome." It is not my intention to suggest a perfect study design is necessary to make any

statements regarding the effect of DNAm in TBRS and the R882 variant specifically. Indeed, the experiments presented are informative and move the field forward. What cannot be done, is drawing broad conclusions from DNA methylation data on n=3 samples on complex disease mechanisms, i.e. suggesting inactivation of an allele.

Reviewer #4 (Remarks to the Author):

In this paper, Smith et al. describe the clinical and molecular phenotypes associated with germline DNMT3A LOF mutations (DOS). They expand on the body of literature related to this syndrome and on the study of DNMT3A mutations in hematological malignancies. The results are largely confirmatory of prior studies. The major novel contributions are the confirmation of a hypomethylation phenotype when there is germline loss of DNMT3A and evidence for orthogonal behavioral, morphometric, and neoplastic phenotypes seen with R882/R878 mutations in humans and mice. The paper also provides some evidence for transcriptional changes seen in the setting of the mutations in whole blood or bone marrow. The major weakness is that there is not much insight into how the mutations lead to the diverse clinical phenotypes in DOS.

Reviewer 1 correctly pointed out in the first round of reviews that drawing conclusions about gene expression changes attributable to the mutations based on a single sample is unwise. Same with the example of the NK and T cell ratios. However, in the authors' defense, they acknowledge that the specific examples they present are not particularly informative about the biology of this syndrome. They do also provide some evidence for orthogonal changes in humans and mice for some of these findings. Overall, I am inclined to allow the authors to make the claims they do about these studies, since the claims overall are modest.

The other studies regarding the phenotype of the Dnmt3a R878H mice are well executed and presented and a nice addition to the field.

Reviewer #2 (Remarks to the Author):

In the revised manuscript the authors provided clear answers to most of my comments. However, before publishing, we recommend that the authors address one concern regarding the analysis of the single RNA seq data. In principle, I agree with the authors that in single cell analysis one sample per group could be representative enough to achieve statistically significant differences since you compare a parameter measured in thousands of cells. For instance, when you compare the mean expression level of a gene in a cell population. In this case each group contains hundreds or thousands of cells (n=100/1000...etc). However, when the authors analyzed the percentage of different cell types per group, for instance the percentage of B-cells in each group, they are measuring one parameter in a group which includes only one sample. Since n=1, the authors can't apply statistical analysis. Therefore, I suggest that the authors change the bar plots in Figure 2b and Supp 2b, by a pie chart as shown in Figure 6b.

As requested, we changed Figure 2b and Supplemental Figure 2c to the pie chart format. We removed p-values for both figures, and amended the text in the manuscript, with p-values and references to significance were removed.

Reviewer #3 (Remarks to the Author):

The authors have implemented many of the suggestion from all three reviewers, which have successfully improved the quality of the manuscript.

For the DNA methylation analysis, the authors have made efforts to implement many of the changes suggested. Particularly, the addition of 9 controls that better match the age distribution of the cases strengthens confidence that the effects seen are due to genotype and not age. I should clarify, that it was not the intention of this reviewer to imply that 33 controls should be used, indeed this is an unreasonable suggestion for this dataset. The suggestions was only meant to contextualize this study in the context of the standards that exist for DNAm microarray experiments to which this study will be compared. The effects of TBRS on DNAm relatively large compared to other disorders of the epigenetic machinery: the effect size is large and many CpG sites are affected, as data here and reported elsewhere show. Therefore, as the authors note, changes made to the experiment design and/or statistical analysis had minimal effects on the data output. Importantly, this is not a reason not to undertake proper statistical and experiment design approaches. WGBS data in epigenetic regulatory disorders are only now becoming common, and this paper will be part of setting these standards. This has been the reason for this reviewer's insistence on certain standards, even if the authors (rightly) feel they may have limited impact on the final data reported.

Specifically, the main issues are age and sex. The authors state that they have accounted for age and sex as covariates, and that this didn't impact the results. Again, this is not unexpected given the effect size. It is noted in the manuscript now that this was done, which is important. However, it is not clear if the data shown throughout are the result of this analysis. If all the DMRs remained the same, why not report the analysis with age and sex as covariates? The same is true for the mouse analysis. It appears that rather than report the new analysis, the original dataset was simply check against the new one with proper covariates used.

We have added the sex of patients in the results sections of the WGBS for human and mouse.

For the human data, the section now reads *"We compared the methylation levels of 11 DOS samples to the peripheral blood samples of 15 healthy donors aged 4-43 years (DNMT3A^{+/+}; 8 male, 7 female). We*

subcategorized DOS patients into two groups: DNMT3A^{R882} (n=3; 1 male, 2 female) and DNMT3A^{non-R882} (n=8; 5 male, 3 female) for subsequent analysis, since each non-R882 patient had a unique mutation.”

For the mouse analyses, we ensured that control and *Dnmt3a*^{R878H/+} mice were matched for sex and age. For mice, the section now reads “To understand the methylation phenotypes of bone marrow cells derived from unmanipulated, littermate-matched *Dnmt3a*^{+/+} (n=10; 2-52 weeks of age, 5 male, 5 female) or *Dnmt3a*^{R878H/+} mice (n=6; 8-38 weeks of age, 3 male, 3 female), we performed WGBS. There was no significant difference in age between control vs. *Dnmt3a*^{R878H/+} groups (p=0.779; two-sample t-test). We also included bone marrow cells from unmanipulated germline *Dnmt3a*^{-/-} mice (n=4; 2 weeks of age, 2 male, 2 female) and *Dnmt3a*^{+/-} mice (n=4; 12-52 weeks of age, 3 male, 1 female) as comparators to calibrate the methylation phenotypes of *Dnmt3a*^{R878H/+} relative to deficiency or haploinsufficiency.”

We determined that there were no significant differences in the ages of the human DOS patients vs. controls, or the R878H mice vs. WT controls.

For the human data, the section now reads “There was no significant difference in age between control vs. DNMT3A^{R882} (p=0.157) or control vs. DNMT3A^{non-R882} (p=0.824) groups (two-sample t-test).”

For the mouse data, the section now reads “There was no significant difference in age between control vs. *Dnmt3a*^{R878H/+} groups (p=0.779; two-sample t-test).”

Using array-based data, Jeffries et al³ reported an “accelerated aging phenotype” in DOS patients, but only one of 353 CpGs from the Horvath “epigenetic clock” study actually overlapped with the differentially methylated CpGs associated with DNMT3A mutations. Further, Jeffries et al used age as a covariate and found that 80% of differentially methylated probes overlapped when age was either included or excluded as a covariate. This strongly suggests that the predominant determinant was genotype, not age.

To identify DMRs with our WGBS studies, we have used the Metilene software, which detects regions that are differentially methylated between 2 groups of samples. Our group has used and tested this tool extensively for the past 5 years, and have found it to work very well for the study of DNMT3A-related methylation phenotypes. Since we agree with the reviewer’s comment that age and sex could be potential confounders, we performed a post-hoc sensitivity analysis, using linear regression to test that, for the regions discovered by Metilene, the genotype remained a significant predictor of methylation beta value, even after accounting for age and sex. We believe this to be a conservative approach, with the benefit that it allows us to use our well documented methods for assessing WGBS data, while also ensuring that our results and conclusions are valid.

To strengthen these points, we added the following section to the discussion:

Because the age and sex of the samples from both species were well matched, it is very unlikely that these DMRs were significantly affected by either covariate. Further, a post-hoc sensitivity analysis of defined DMRs using linear regression to test for age and sex as covariates revealed that genotype remained the significant predictor of methylation phenotype.

Reviewer 3 acknowledges that it is highly unlikely that inclusion of covariates in the DMR calling itself will impact the results of our analysis in any relevant way. He/she, however, would still like us to incorporate those covariates for the sake of setting a standard for future methylation studies, in general. If it were the case that we were examining, for instance, common variants having subtle phenotypic effects in a large population, we would absolutely agree with this approach, and we would have chosen and tested a different analytical approach from the outset. As it stands, however, we are studying genetic variants with a very large effect on DNA methylation, and we have chosen our methods accordingly.

There do exist some bioinformatic tools (e.g., DMRSeq) which purport to incorporate covariates into the DMR calling step. Our group has not tested these relatively new methods, and we do not have the years of experience with them as we do with Metilene; there is no guarantee that they will work well, or provide reliable results with WGBS data. Although it would be possible for us to use alternative analysis pipelines

and redo all analyses, this would not necessarily set a better standard for this work, given that these tools are not as well vetted as Metiline for WGBS data.

The author's response regarding cell composition is sufficient given the limitations of WGBS at this time. Noting that cell types are not reported to change is the best that can be done currently. However, the author's response that: "Furthermore, WGBS is different from array-based methylation methodologies because the entire CpG space in the genome is covered without bias." is confusing, it is unclear what relevance the difference between DNAm microarray and WGBS technology has here.

The two previous studies of methylation patterns in the peripheral blood of DOS patients have been based on methylation arrays. Although these arrays are inexpensive and expedient, they are biased toward specific annotated regions of the genome, and the standard 450K arrays cover <2% of the 28 million CpGs in the human genome. WGBS is completely unbiased, and with the level of coverage we obtained in this study, the methylation status of more than 98% of CpGs are measured accurately. Since nearly 50 times the number of CpGs is assessed with WGBS, this method provides more accurate and more comprehensive information about the methylomes of these patients, and allowing us to more precisely define the regions where DNMT3A acts in the genome.

This issue is whether differences in DNAm between case groups (detected using either method) are in part due to differences in changing cell populations in the case group, not due to actual differences in DNAm in those cells. Again, this cannot be addressed yet in WGBS data, as such it should be listed as a limitation.

There are two parts to this answer:

First, for the human datasets, all the patients had normal blood counts at the times that the peripheral blood samples were drawn for the study. For the single R882H patient where we had fresh material available (from him and his normal brother), both scRNA-seq (Figure 2a, b, Supp Figure 2a,b) and a careful flow cytometry analysis (Supp Fig 2c) revealed only small differences in the cellular compositions of the samples. Likewise, for the WT vs R878H mice, scRNA-seq data shown in Figure 5a,b, and Supp Figure 5a,b, c show minimally perturbed cellular populations in mouse bone marrow. In Supp Figure 7, we show flow data from the peripheral blood of 17 WT vs. 23 R878H mice, and bone marrow from 31 WT vs. 29 R878H mice at ages ranging from <100 to >200 days of age, demonstrating the very small differences in cellular populations in the blood and marrow samples of these mice. Since the DNA content of all cell types is equivalent, the differences caused by altered populations of cells would be proportional to the change in size of the compartment. Since the cellular changes present in these samples are small, they cannot account for the large effect sizes of DNA methylation seen in these studies.

Secondly, multiple lines of evidence have previously shown that differing cell populations cannot account for DNMT3A-dependent methylation phenotypes associated with loss-of-function mutations in *DNMT3A*.

1. In Spencer et al (Cell, 2017)¹ from this lab, the purified neutrophils, monocytes and T-cells from TBRS1 and his normal brother were sorted and subjected to WGBS. The DMRs detected in these three cell types that dominate the peripheral blood (accounting for over 90% of cells) had nearly identical phenotypes in all three populations.

2. In Ketkar et al (PNAS 2020)², we showed that in non-leukemic murine hematopoietic cells with Dnmt3a deficiency, the DMRs associated with Dnmt3a deficiency in purified stem, progenitor and mature lineage cells are essentially equivalent.
3. The array-based study of Jeffries et al³ likewise noted “that the observed patterns of differential DNA methylation are not strongly influenced by cell-type variation”, using inferred blood cell composition from DNA methylation-based estimates.
4. Finally, the publication by Aref-Eshghi⁴ evaluated the extent to which the variation in blood cell type compositions influenced the episignatures identified in their patient cohorts. They found less than 5% inter-cell type variability for all 34 episignature classes.

The authors efforts to scale back the language around interpretation around of the difference in DNAm severity between the non-R882 and R882 variants is appreciated. The authors respond, “As the reviewer suggests, a perfect study would include multiple samples from a worldwide cohort of DOS patients with sex and age-similar unaffected siblings. We do not have the resources to perform such a study. Until this can be done, smaller studies that evaluate as many DOS patients as possible will provide important insights that can still have an impact on our understanding of this syndrome.” It is not my intention to suggest a perfect study design is necessary to make any statements regarding the effect of DNAm in TBRS and the R882 variant specifically. Indeed, the experiments presented are informative and move the field forward. What cannot be done, is drawing broad conclusions from DNA methylation data on n=3 samples on complex disease mechanisms, i.e. suggesting inactivation of an allele.

One important clarification should be made here: the R882 mutations in DNMT3A are very well established biochemically and functionally as having dominant negative activity in studies from our lab and many others. In this study, all three R882 patients had near identical DMRs compared to healthy donors. One novel question for this study was the relative methylation phenotypes for the R882 vs. non-R882 mutations, which are far more heterogeneous, and for which the mechanisms have not been established. These comparisons were not described in the previous array-based studies of DOS patients.

One of the non-R882 patients in this study serendipitously had a clean deletion of *DNMT3A*, which ‘calibrates’ the phenotype of haploinsufficiency, allowing us to determine whether the other 7 non-R882 patients were more similar to this patient, or to the dominant negative R882 patients. Clearly, the other 7 non-R882 patients had methylation phenotypes more similar to the haploinsufficient patient, allowing us to speculate that these mutations may all cause inactivation of the mutant allele.

We do not have biochemical evidence to prove this hypothesis, however, and have therefore amended the discussion as follows: “*By coincidence, one patient had a true haploinsufficient state (UPN 518693), due to a heterozygous ~135 kb deletion that encompassed the entire DNMT3A gene (Table 1, Supplementary Fig 1). This patient’s methylation phenotype defined the consequences of simple haploinsufficiency due to gene deletion. Since all of the other non-R882 missense mutations had a similar magnitude of methylation loss (Fig 1), we suggest that the non-R882 mutations in this study may all lead to functional deficits that mimic inactivation of the affected allele. However, the mechanisms of inactivation are almost certainly multifarious.*”

Finally, while our focus has been on genotype-dependent changes in the methylome, we certainly agree with the reviewer that others may be very interested in age and sex and the way these impact methylation phenotypes in hematopoietic cells. All of our data is therefore available on dbGaP and the SRA for more detailed studies with new algorithms designed to evaluate these and/or additional covariates.

1. Spencer, D.H. *et al.* CpG Island Hypermethylation Mediated by DNMT3A Is a Consequence of AML Progression. *Cell* **168**, 801-816 e13 (2017).

2. Ketkar, S. *et al.* Remethylation of Dnmt3a (-/-) hematopoietic cells is associated with partial correction of gene dysregulation and reduced myeloid skewing. *Proc Natl Acad Sci U S A* **117**, 3123-3134 (2020).
3. Jeffries, A.R. *et al.* Growth disrupting mutations in epigenetic regulatory molecules are associated with abnormalities of epigenetic aging. *Genome Res* **29**, 1057-1066 (2019).
4. Aref-Eshghi, E. *et al.* Evaluation of DNA Methylation Episignatures for Diagnosis and Phenotype Correlations in 42 Mendelian Neurodevelopmental Disorders. *Am J Hum Genet* **106**, 356-370 (2020).

REVIEWERS' COMMENTS

Reviewer #2 (Remarks to the Author):

the authors have adequately addressed all my concerns. I believe that the manuscript is now fit for publication.

Reviewer #3 (Remarks to the Author):

We are satisfied with the responses provided by the authors.